# The CENP-A centromere targeting domain facilitates H4K20 monomethylation in the nucleosome by structural polymorphism

Yasuhiro Arimura [1,2], Hiroaki Tachiwana[2,3], Hiroki Takagi[1,2], Tetsuya Hori[4], Hiroshi Kimura [5], Tatsuo Fukagawa [4] & Hitoshi Kurumizaka [1,2]

Centromeric nucleosomes are composed of the centromere-specific histone H3 variant CENP-A and the core histones H2A, H2B, and H4. To establish a functional kinetochore, histone H4 lysine-20 (H4K20) must be monomethylated, but the underlying mechanism has remained enigmatic. To provide structural insights into H4K20 methylation, we here solve the crystal structure of a nucleosome containing an H3.1-CENP-A chimera, H3.1$^{CATD}$, which has a CENP-A centromere targeting domain and preserves essential CENP-A functions in vivo. Compared to the canonical H3.1 nucleosome, the H3.1$^{CATD}$ nucleosome exhibits conformational changes in the H4 N-terminal tail leading to a relocation of H4K20. In particular, the H4 N-terminal tail interacts with glutamine-76 and aspartate-77 of canonical H3.1 while these interactions are cancelled in the presence of the CENP-A-specific residues valine-76 and lysine-77. Mutations of valine-76 and lysine-77 impair H4K20 monomethylation both in vitro and in vivo. These findings suggest that a CENP-A-mediated structural polymorphism may explain the preferential H4K20 monomethylation in centromeric nucleosomes.

[1] Laboratory of Chromatin Structure and Function, Institute for Quantitative Biosciences, The University of Tokyo, 1-1-1 Yayoi, Bunkyo-ku, Tokyo 113-0032, Japan. [2] Graduate School of Advanced Science and Engineering, Waseda University, 2-2 Wakamatsu-cho, Shinjuku-ku, Tokyo 162-8480, Japan. [3] The Cancer Institute of Japanese Foundation for Cancer Research, 3-8-31 Ariake, Koto-ku, Tokyo 135-8550, Japan. [4] Graduate School of Frontier Biosciences, Osaka University, Suita, Osaka 565-0871, Japan. [5] Cell Biology Center, Institute of Innovative Research, Tokyo Institute of Technology, 4259 Nagatsuta-cho, Midori-ku, Yokohama 226-8501, Japan. These authors contributed equally: Yasuhiro Arimura, Hiroaki Tachiwana, Hiroki Takagi. Correspondence and requests for materials should be addressed to H.K. (email: kurumizaka@iam.u-tokyo.ac.jp)

Accurate chromosome segregation during mitosis is mediated by the attachment of spindle microtubules to the kinetochore, which is formed on the centromere of each chromosome[1,2]. Therefore, correct centromere formation and inheritance are crucial for accurate chromosome segregation. For these processes, the centromere must be formed in the specific region on a chromosome. In most eukaryotes, the centromere is specified by DNA sequence-independent epigenetic mechanisms, and the centromere-specific histone H3 variant, CENP-A, plays a critical role as a key epigenetic marker for centromere specification[3–8]. CENP-A is a protein that specifically accumulates on centromeres[9,10] and is homologous to histone H3[11]. CENP-A forms the octameric nucleosome with the core histones H2A, H2B, and H4, as revealed by the crystal structure[12], and creates a foundation to establish centromeric chromatin with the coordination of additional centromere proteins, such as CENP-C[4,13–16], CENP-N[13,17–20], and the Mis18 complex[21,22].

For the CENP-A deposition process, CENP-A modifications, including phosphorylation and ubiquitylation, are considered to facilitate proper CENP-A deposition[23,24], although controversial results have been reported[25]. Acetylation of histone H4 in the CENP-A-H4 pre-deposition complex was also reported[26]. In addition to the modifications of the CENP-A-H4 pre-deposition complex, the histones in the nucleosome containing CENP-A are also modified[27,28]. We previously demonstrated that the histone H4 K20 residue (H4K20) in the CENP-A nucleosome is substantially monomethylated in human and chicken cells, and revealed that this methylation is crucial for kinetochore assembly[28]. As H4K20 also exists in the canonical H3 nucleosome, a critical question is how this modification becomes highly accumulated in the CENP-A nucleosomes at centromeres. It is possible that a methyltransferase for monomethylation, such as PR-Set7, may associate with centromere proteins, but we did not observe the clear centromere localization of PR-Set7[28]. As another possibility, in the CENP-A nucleosome, the H4 N-terminal tail containing the K20 residue may have a certain structural feature that allows-specific monomethylation at the H4K20 residue. However, the H4 N-terminal tail conformation around the H4K20 residue has not been visualized in the crystal structure of the CENP-A nucleosome, because of its insufficient resolution[12].

To visualize the H4 N-terminal tail more clearly in the nucleosome, in this study, we used a chimeric H3.1 containing the CENP-A centromere targeting domain (CATD) region of CENP-A, called H3.1$^{CATD}$, for the structure analysis, instead of the CENP-A nucleosome. The CATD, which is mapped to the CENP-A region containing L1 and the α2 helix, has been identified as the region required for the centromere localization of CENP-A[29,30], and it binds to the CENP-A chaperones, yeast Scm3[31–34] and mammalian HJURP[35–37], in the CENP-A-H4 pre-deposition complex for proper centromere localization[38–40]. The chimeric H3$^{CATD}$ is recruited to centromeres, and partially restores the CENP-A function in CENP-A depleted cells[30,41]. Therefore, we believe that the CATD sequence conserves a critical function for the CENP-A-mediated centromere formation in cells.

Here, we report the crystal structure of the H3.1$^{CATD}$ nucleosome at 2.73 Å resolution. In the structure, the H4 N-terminal tail of the H3.1$^{CATD}$ nucleosome conformation is clearly different from that in the H3.1 nucleosome. The H4 N-terminal tail is released from the H3 molecule in the H3.1$^{CATD}$ nucleosome (the outward H4-N conformation), while it is captured in the H3.1 nucleosome through interactions with Q76 and D77 of H3.1 (the inward H4-N conformation). The H4K20 residue in the CENP-A and H3.1$^{CATD}$ nucleosomes is highly monomethylated, as compared to that in the canonical H3.1 nucleosome.

Consistently, the accumulation of H4K20 monomethylation around the centromeres is significantly decreased in chicken DT40 cells harboring the CENP-A$^{QD}$ mutation, which allows the H4 N-terminal tail to be re-captured by the CENP-A$^{QD}$ mutant in the CENP-A nucleosome. Therefore, we propose that the CENP-A-mediated structural polymorphism of the H4 N-terminal tail is the structural basis for the specific H4K20 monomethylation in the CENP-A nucleosome. More generally, this suggests that one histone variant can allosterically specify the modification of another histone in the nucleosome.

## Results

**Crystal structure of the H3.1$^{CATD}$ nucleosome.** We purified H3.1$^{CATD}$, in which amino acid residues 75–112 of human histone H3.1 were replaced with the corresponding amino acid residues 75–114 of human CENP-A, as a recombinant protein (Fig. 1a). We then reconstituted the nucleosome containing H3.1$^{CATD}$. The purified H3.1$^{CATD}$ nucleosome was crystallized, and its structure was determined at 2.73 Å resolution (Fig. 1b, Supplementary Table 1). The overall structure of the H3.1$^{CATD}$ nucleosome is quite similar to those of other human conventional nucleosomes[42–44]. In the canonical H3.1 nucleosome structure, one of the two H4 N-terminal tails is clearly visible (Fig. 1c, Supplementary Figure 1a). Surprisingly, we found that the orientation of the H4 N-terminal tail in the H3.1$^{CATD}$ nucleosome was substantially different from that observed in the canonical H3.1 nucleosome (Fig. 1d, Supplementary Figures 1, 2, and 3). Therefore, we named the H4 N-terminal tail conformation found in the H3.1$^{CATD}$ nucleosome as the outward H4-N conformation, and that observed in the H3.1 nucleosome as the inward H4-N conformation (Fig. 1d). The corresponding H4 N-terminal tail was not visible in the crystal structure of the CENP-A nucleosome due to its low resolution[12], however, observations of the CENP-A nucleosome structure by cryo-electron microscopy (EM) clearly showed the outward H4-N conformation (Supplementary Figure 4)[18,19]. Interestingly, the canonical H3.1 nucleosome may form both the inward and outward H4-N conformations, because the weak electron density similar to the outward H4-N conformation was observed on the ambiguous side of the H4 N-terminal tail in the H3.1 nucleosome (Supplementary Figures 1 and 5)[42,43,45–47]. However, as the inward H4-N conformation was not observed in the H3.1$^{CATD}$ nucleosome (Supplementary Figure 1 and 2), we propose that the CENP-A CATD residues specifically promote the formation of the outward H4-N conformation.

**CATD V76 and K77 abrogate the local H3-H4 interaction.** In the H3.1 nucleosome structure, the H3.1 Q76 and D77 residues probably interact with the H4 R19 (side chain) and L22 (backbone) residues, respectively, and may restrict the H4 N-terminal tail orientation, forming the inward H4-N conformation (Fig. 2a and Supplementary Figure 6). Interestingly, in the canonical H3.1 nucleosome, the H4 R19 residue directly binds to the DNA backbone, and this H4 R19-DNA binding may be supported by the H3 Q76-H4 R19 interaction (Fig. 2a and Supplementary Figure 6). The interaction between the H3.1 D77 and H4 L22 residues may not be strong, because it is mediated by the Mn$^{2+}$-coordinated water molecule at a distance of 3.4 Å (Fig. 2a). These H3.1 Q76 and D77 residues correspond to the V76 and K77 residues in the CATD. In contrast to Q76 and D77 in the H3.1 nucleosome, the CATD V76 and K77 residues did not interact with the H4 tail, and the H3 Q76-mediated H4 R19-DNA binding was also abolished in the H3.1$^{CATD}$ nucleosome structure (Fig. 2b). Consequently, the H4 N-terminal tail was located farther away from these CATD residues, and was exposed to the solvent (Fig. 2b).

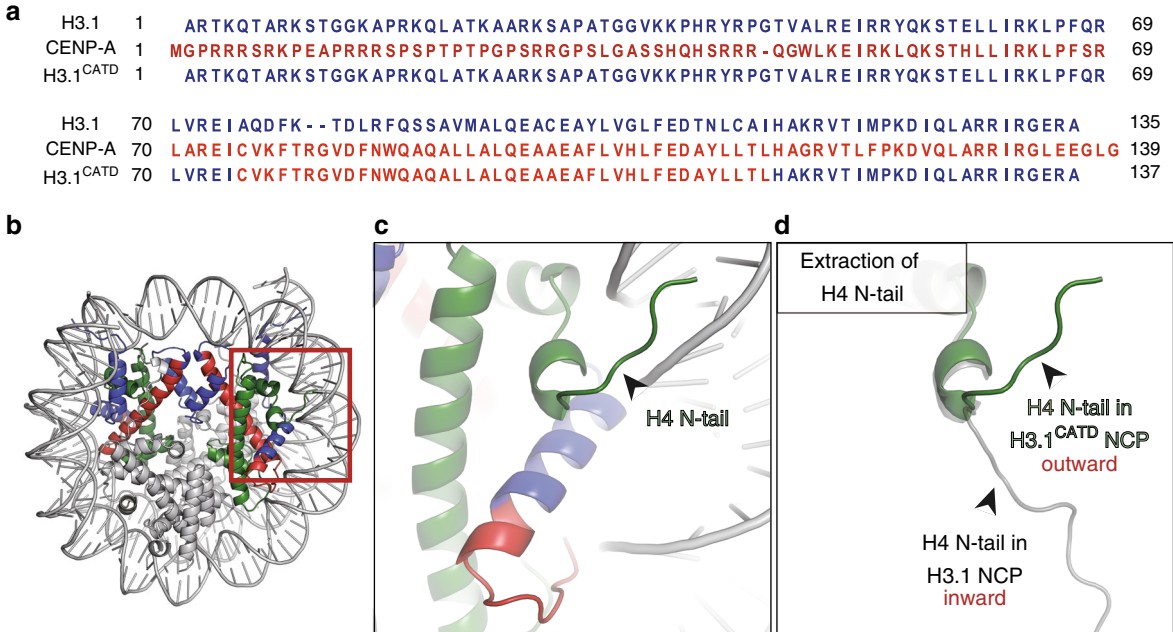

**Fig. 1** Crystal structure of the nucleosome containing H3.1^CATD. **a** Alignment of the amino acid sequences of human histone H3.1, CENP-A, and H3.1^CATD. Amino acid residues derived from H3.1 and CENP-A are colored blue and red, respectively. **b** The crystal structure of the nucleosome containing H3.1^CATD. The H3.1^CATD regions derived from H3.1 and CENP-A are colored blue and red, respectively. H4 is colored green. H2A, H2B, and DNA are colored gray. **c** Close-up view of the H4 N-terminal tail in the H3.1^CATD nucleosome. The region enclosed by the red rectangle in **b** is enlarged. **d** Structural comparison of the H4 N-terminal tails between the H3.1^CATD nucleosome and the H3.1 nucleosome[76] (PDB ID: 5Y0C). The H4 N-terminal tails in the H3.1^CATD and H3.1 nucleosomes are colored green and gray, respectively

To determine whether the H3.1 Q76 and D77 residues actually define the H4 N-terminal tail orientation in the nucleosome, we prepared the H3.1^CATD(V76Q K77D) mutant (Fig. 2c). We then determined the crystal structure of the nucleosome containing H3.1^CATD(V76Q K77D) at 2.58 Å resolution (Fig. 2d). In the H3.1^CATD(V76Q, K77D) nucleosome structure, the Q76 and D77 residues, replacing the corresponding V76 and K77 residues of the H3.1^CATD, interact with the H4 R19 and L22 residues, accompanied by the H4 R19-DNA binding, as observed in the canonical H3.1 (Fig. 2e and Supplementary Figure 6). As a result, the H4 N-terminal tail conformation in the H3.1^CATD(V76Q, K77D) nucleosome adopted the inward H4-N conformation, as observed in the canonical H3.1 nucleosome (Fig. 2f). In both the canonical and H3.1^CATD(V76Q, K77D) nucleosomes, the visible H4 tail interacted with the acidic patch of the neighboring nucleosome in the crystals, and formed the inward configuration. In contrast, in the H3.1^CATD nucleosome, the H4 tails in the outward configuration may not interact with the acidic patch of the neighboring nucleosome. These structural findings indicated that the H3.1 Q76 and D77 residues actually function to constrain the H4 N-terminal tail orientation in the nucleosome. As we previously demonstrated that H4K20 in the N-terminal tail is highly monomethylated in the CENP-A nucleosome, as compared to that of the H3 nucleosome in centromeres[28], we hypothesized that the CATD-dependent abrogation of the local H3 and H4 N-terminal tail interactions, mediated by the CATD V76 and K77 residues, could be the structural basis for the specific H4K20 monomethylation in the CENP-A nucleosome (Fig. 3a).

**CATD stimulates H4K20 monomethylation in the nucleosome.** To directly test our hypothesis that the CATD V76 and K77 residues facilitate the hyper monomethylation of the H4K20

residue in the CENP-A nucleosome, we prepared the CENP-A^QD mutant, in which the human CENP-A V76 and K77 residues are replaced with the corresponding H3.1 Q76 and D77 residues (Fig. 3b). We then reconstituted the nucleosomes containing canonical H3.1, CENP-A, H3.1^CATD, and the CENP-A^QD mutant (Supplementary Figure 7a, b), and performed the H4K20 monomethylation assay in vitro (Fig. 3c). As PR-Set7 is the enzyme responsible for the H4K20 monomethylation in the nucleosome[28,48], human PR-Set7 was employed for this methylation assay (Supplementary Figure 7c). The reconstituted nucleosomes were incubated with human PR-Set7 in the presence of S-adenosylmethionine, and the resulting K20 monomethylated H4 was detected by western blotting, using a monoclonal antibody against monomethylated H4K20[49]. As an internal control, H2B was simultaneously detected, using an anti-H2B antibody.

We then performed time course experiments to monitor the methyltransferase activity of PR-Set7 with nucleosome templates. As anticipated, the H4K20 residue of the CENP-A nucleosome was substantially monomethylated, as compared to that of the H3.1 nucleosome (Fig. 3d and Supplementary Figure 8a). Similar results were obtained with the H3.1^CATD nucleosome (Fig. 3e and Supplementary Figure 8b). Therefore, PR-Set7 preferentially monomethylates the nucleosomes containing CATD. Strikingly, the H4K20 monomethylation in the CENP-A nucleosome was substantially suppressed in the CENP-A^QD mutant nucleosome (Fig. 3f and Supplementary Figure 8c). These results strongly support our hypothesis that the CENP-A V76 and K77 residues cancel the constraint of the H4 N-terminal tail, and facilitate the specific monomethylation of the H4K20 residue in the CENP-A nucleosome.

We also performed a trypsin proteinase assay to evaluate the flexibility of the H4 N-terminal tails in the nucleosome, which revealed that the H4 N-terminal tail flexibility was similar in both

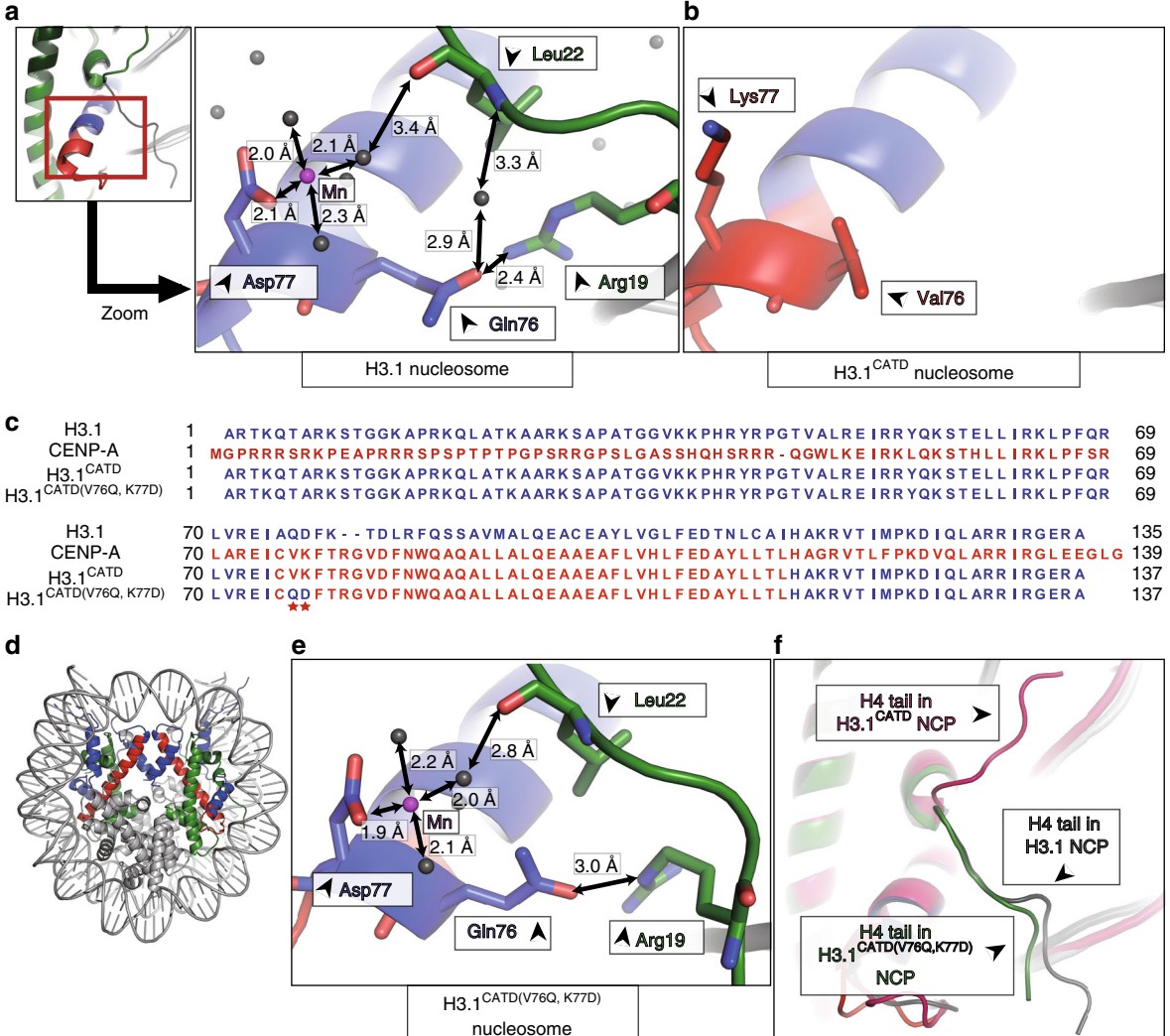

**Fig. 2** The human CATD residues Val76 and Lys77 do not bind to the H4 N-terminal tail. **a** Close-up view of the region around the positions of the Q76 and D77 residues of H3.1 in the nucleosome[76] (PDB ID: 5Y0C). The H3.1 Q76 and D77 residues interact with the H4 R19 (side chain) and L22 (backbone) residues, respectively. Possible hydrogen bonds are presented as arrows. **b** Close-up view of the region around the positions of the V76 and K77 residues of CATD in the nucleosome. The CATD V76 and K77 residues do not interact with the H4 residues. **c** Alignment of the amino acid sequences of human histone H3.1, CENP-A, H3.1$^{CATD}$, and H3.1$^{CATD(V76Q, K77D)}$. Amino acid residues derived from H3.1 and CENP-A are colored blue and red, respectively. Stars indicate the positions of the CATD V76 and K77 residues. **d** The crystal structure of the nucleosome containing the H3.1$^{CATD(V76Q, K77D)}$. The H3.1$^{CATD(V76Q, K77D)}$ regions derived from H3.1 and CENP-A are colored blue and red, respectively. H4 is colored green. H2A, H2B, and DNA are colored gray. **e** Close-up view of the region around the positions of the Q76 and D77 residues of H3.1$^{CATD(V76Q, K77D)}$ in the H3.1$^{CATD(V76Q, K77D)}$ nucleosome. The H3.1$^{CATD(V76Q, K77D)}$ Q76 and D77 residues interact with the H4 R19 (side chain) and L22 (backbone) residues, respectively. Possible hydrogen bonds are presented as arrows. **f** Structural comparison of the H4 N-terminal tails between the H3.1$^{CATD(V76Q, K77D)}$ nucleosome, the H3.1$^{CATD}$ nucleosome, and the H3.1 nucleosome[76] (PDB ID: 5Y0C). The H4 N-terminal tails in the H3.1$^{CATD(V76Q, K77D)}$ nucleosome, the H3.1$^{CATD}$ nucleosome, and the H3.1 nucleosome are colored green, pink, and gray, respectively

the CENP-A and H3.1 nucleosomes (Fig. 3g, Supplementary Figure 9b, and Supplementary Figure 11). The binding affinity of PR-Set7 to the CENP-A and H3.1 nucleosomes was similar (Fig. 3h and Supplementary Figure 9a). These data provide additional evidence that the H4 N-terminal tail flexibility and the PR-Set7 binding efficiency are not the direct reasons for the enhanced H4K20 monomethylation in the CENP-A nucleosome. Therefore, the outward H4-N conformation predominantly formed in the CENP-A nucleosome may be important for efficient H4K20 monomethylation.

**CENP-A$^{QD}$ reduces the centromeric H4K20 monomethylation.** We next tested the effect of the CENP-A$^{QD}$ mutation on the CENP-A nucleosome-specific H4K20 monomethylation in vivo.

For this experiment, we utilized chicken DT40 cells, because we previously detected the accumulation of H4K20 monomethylation in chicken centromeres, some of which have unique sequences identifiable by sequencing coupled with chromatin immunoprecipitation (ChIP-seq). As the human CENP-A V76 and K77 residues correspond to the L67 and L68 residues in chicken CENP-A, we mutated the L67 residue to Q and the L68 residue to D, and made the chicken CENP-A$^{QD}$ construct (Fig. 4a). The GFP-tagged chicken CENP-A$^{QD}$ or wild-type chicken CENP-A was expressed in CENP-A-deficient DT40 cells, and the CENP-A was completely replaced with GFP-tagged CENP-A$^{QD}$ or wild-type CENP-A in these cell lines (Fig. 4b, c and Supplementary Figure 10). Using these cells, we evaluated the accumulation of the H4K20 monomethylation around the

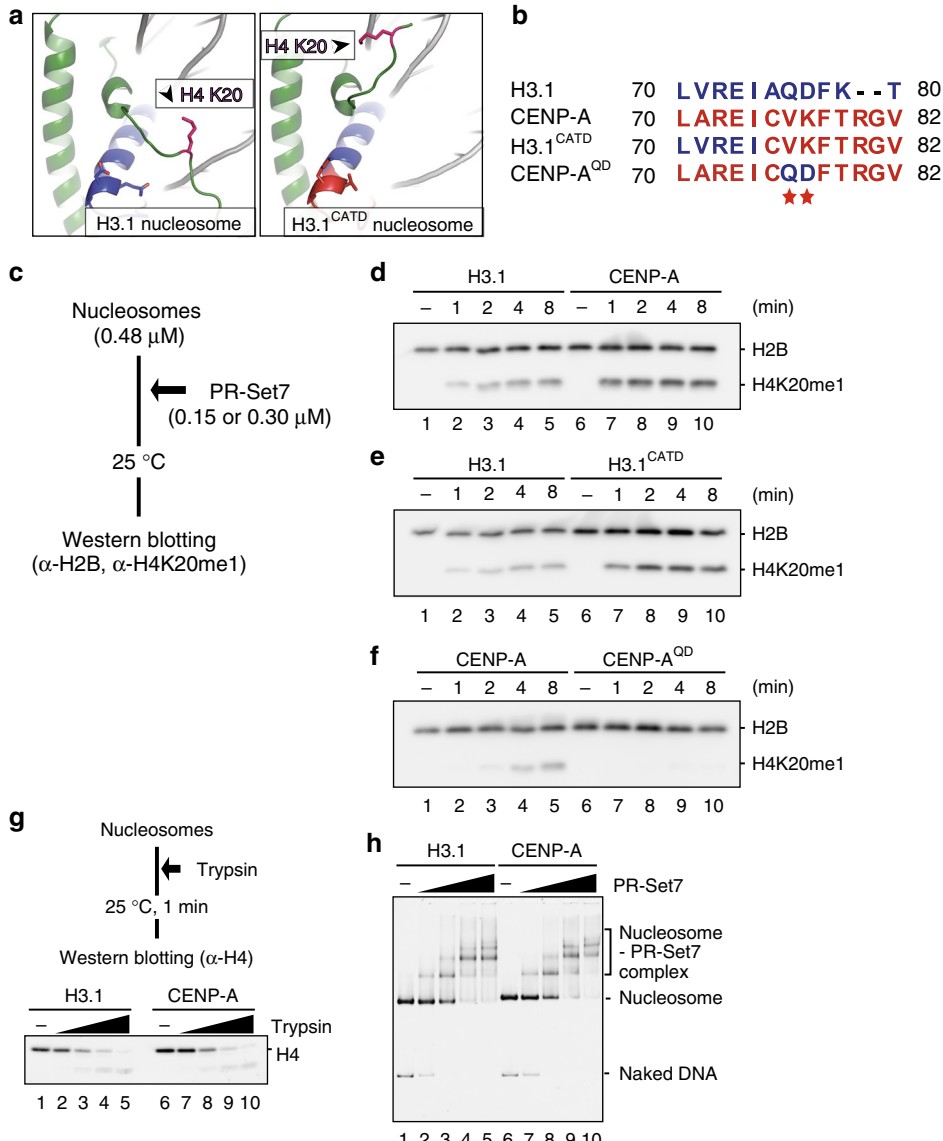

**Fig. 3** H4K20 monomethylation efficiently occurs in CENP-A and H3.1$^{CATD}$ nucleosomes. **a** Structural comparison near the H4 K20 residue between the H3.1$^{CATD}$ nucleosome and H3.1 nucleosome[76] (PDB ID: 5Y0C). The side chain of the H4 K20 residue is shown in purple for each nucleosome. **b** Alignment of the amino acid sequences of human H3.1, CENP-A, H3.1$^{CATD}$, and CENP-A$^{QD}$. **c** Experimental scheme for the in vitro methyltransferase assay. **d–f** Representative western blotting data of the time course analysis of the methyltransferase assay with the H3.1 and CENP-A nucleosomes. Experiments were repeated at least three times, and the reproducibility was confirmed (Supplementary Figure 8). **d** Lanes 1–5 and 6–10 indicate results for the H3.1 and CENP-A nucleosomes, respectively. The purified nucleosome (0.48 μM) containing H3.1 or CENP-A was incubated with His$_6$-tagged PR-Set7 (0.30 μM). **e** Lanes 1–5 and 6–10 indicate results for the H3.1 and H3.1$^{CATD}$ nucleosomes, respectively. The purified nucleosome (0.48 μM) containing H3.1 or CENP-A was incubated with His$_6$-tagged PR-Set7 (0.30 μM). **f** Lanes 1–5 and 6–10 indicate results for the CENP-A and CENP-A$^{QD}$ nucleosomes, respectively. The purified nucleosome (0.48 μM) containing CENP-A or CENP-A$^{QD}$ was incubated with His$_6$-tagged PR-Set7 (0.15 μM). **g** Experimental scheme for the proteinase accessibility assay with trypsin (Upper panel) and representative western blotting data (Lower panel). Lanes 1–5 and 6–10 indicate results for the H3.1 nucleosome and the CENP-A nucleosome, respectively. Three-independent experiments were performed. The data of replicated experiments and the full image are shown in Supplementary Figures 9 and 11, respectively. **h** Gel shift assays of the H3.1 nucleosome and the CENP-A nucleosome with or without His$_6$-tagged PR-Set7. Lanes 1–5 and 6–10 indicate the results for the H3.1 nucleosome and the CENP-A nucleosome, respectively. The gel was stained with SYBR-Gold. Three-independent experiments were performed, and the reproducibility was confirmed (Supplementary Figure 9)

centromere in chromosome Z by spike-in ChIP-seq experiments with anti-H4K20me1 and anti-*Drosophila* H2Av antibodies (Fig. 4d)[50,51].

In these experiments, a fixed amount of a chromatin sample from *Drosophila melanogaster* S2 cells was added to our experimental samples from chicken DT40 cells. We then performed a ChIP-seq analysis with a target antibody (either anti-GFP or anti-H4K20me1) and an anti-*Drosophila* H2Av antibody. We mapped the sequence data into the chicken and

*Drosophila* genome databases, and the sequence reads in the chicken genome were normalized with the read-counts mapped to the *Drosophila* genome. We used two-independent CENP-A knockout chicken DT40 cell lines expressing GFP-CENP-A$^{QD}$ (#2-5 and #3-1). The expression level of GFP-CENP-A$^{QD}$ in the #2-5 clone was similar to that of GFP-CENP-A in cells expressing GFP-CENP-A, and the GFP-CENP-A$^{QD}$ expression level in the #3-1 clone was slightly higher than that in the #2-5 clone (Fig. 4c and Supplementary Figures 10, 12, and 13). The H4K20

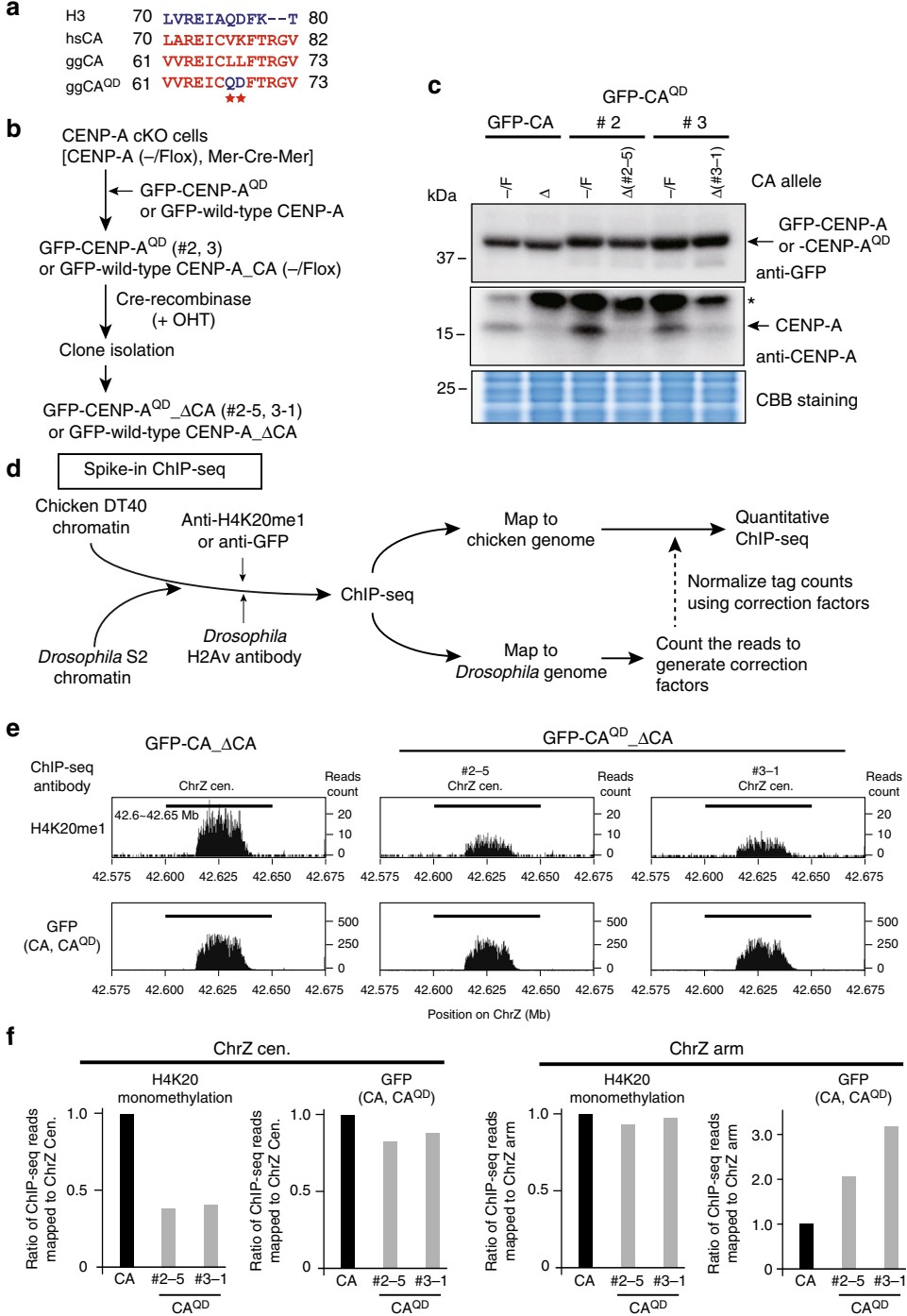

**Fig. 4** CENP-A$^{QD}$ mutation causes reduced H4K20 monomethylation in the DT40 centromere. **a** Alignment of the amino acid sequences of human histone H3.1, human CENP-A (hsCA), chicken CENP-A (ggCA), and chicken CENP-A$^{QD}$ (ggCA$^{QD}$). The amino acid residues corresponding to the human CENP-A V76 and K77 residues are the chicken CENP-A L67 and L68 residues, respectively. **b** Experimental design to create CENP-A-deficient cells expressing either GFP-tagged CENP-A or CENP-A$^{QD}$. **c** Immunoblot analyses with anti-GFP (Upper panel) and anti-CENP-A (Middle panel) antibodies, using whole-cell protein extracts from cells expressing GFP-tagged CENP-A (GFP-CA) or CENP-A$^{QD}$ before (-/F) and after (Δ) the endogenous CENP-A knockout. Two-independent cell lines were used for cells expressing CENP-A$^{QD}$ [#2 and #3]. Endogenous CENP-A depletion was confirmed (Middle panel). The asterisk indicates non-specific bands. Coomassie Brilliant Blue (CBB) staining is shown as a loading control (Lower panel). **d** Schematic representation of the Spike-in ChIP-seq. **e** Normalized ChIP-seq profiles with anti-H4K20me1 (Upper panels) and anti-GFP (Lower panels) around the centromere region on chromosome Z (42.575–42.675 Mb) in CENP-A-deficient DT40 cell lines, expressing either GFP-CENP-A (Left side panels) or GFP-CENP-A$^{QD}$ (#2-5; Middle and #3-1; Right panels). The centromeric accumulation of H4K20me1 was reduced in the cells expressing GFP-CENP-A$^{QD}$ (Upper middle and right panels). **f** Ratio of normalized ChIP-seq reads mapped to the chromosome Z centromere (ChrZ cen.: 42.6–42.65 Mb, Left two graphs). Ratios of reads mapped to the non-centromeric chromosome Z arm are also shown (ChrZ arm, Right two graphs). The reads of H4K20me1 mapped to the centromeric region in the GFP-CENP-A$^{QD}$ #2-5 and #3-1 cell lines were reduced to 0.38 and 0.41, respectively, as compared to the reads in cells expressing GFP-CENP-A. The mapped reads to the arm region were similar in both cells expressing GFP-CENP-A and GFP-CENP-A$^{QD}$

monomethylation levels around the centromeric region ware substantially lower in both the #2-5 and #3-1 cell lines (~40%) than those in the cells expressing GFP-CENP-A, but the levels in the non-centromeric regions were not affected (Fig. 4e and f). However, the centromere accumulation of GFP-CENP-A[QD] (#2-5 and #3-1) was slightly reduced (~80% level), as compared to that of GFP-CENP-A (Fig. 4d, e and f). The non-centromeric CENP-A was increased in the #2-5 and #3-1 clones, probably due to the excess production of GFP-CENP-A[QD] (Fig. 4c, e and f). Based on these results obtained by the quantitative spike-in ChIP-seq analyses using two-independent chicken DT40 clones, we conclude that the absence of the interaction of the H4 N-terminal tail with CENP-A in the nucleosome facilitates the efficient centromeric H4K20 monomethylation in vivo.

## Discussion

We previously demonstrated that H4K20 monomethylation substantially accumulates in the nucleosomes containing CENP-A in human and chicken centromeres[28]. A reduction of the H4K20 monomethylation level, by tethering the PHF8 demethylase in centromeres, caused defects in kinetochore assembly and mitotic progression in the cells[28]. These results suggest that the H4K20 monomethylation in the CENP-A nucleosome is a key step to establish the functional kinetochore. However, the mechanism by which the H4K20 monomethylation preferentially occurs in the CENP-A nucleosome has remained elusive. In this report, based on structural analyses combined with in vitro and in vivo assays for H4K20 monomethylation, we conclude that the absence of the local interaction between CENP-A and the H4 N-terminal tail predominantly forms the outward H4-N conformation, and preferentially facilitates the H4K20 monomethylation in the CENP-A nucleosome, as compared to the H3.1 nucleosome.

Many crystal structures of nucleosomes have been reported[52]. Interestingly, in several cases, only one of the two H4 N-terminal tails adopts a similar conformation to that in the H3.1[CATD] nucleosome (outward H4-N conformation, Supplementary Figure 5). To form the inward H4-N conformation observed in the H3.1 nucleosome, the interactions between the H3.1 Q76 and D77 residues and the H4 R19 and L22 residues are essential (Fig. 2a). Since these H3-H4 interactions do not occur in CENP-A, the inward H4-N conformation in the CENP-A nucleosome cannot be formed. In contrast, in the H3 nucleosome, the two H4 tails can form both the inward and outward H4-N conformations. It is plausible that the H4K20 monomethylation preferentially occurs in the CENP-A nucleosome, if the outward H4-N conformation is more suitable for the methyltransferase reaction than the inward H4-N conformation.

Recent structural studies have suggested that either CENP-C or CENP-N directly recognizes the CENP-A nucleosome, to facilitate kinetochore assembly[15,18–20]. In addition to the direct binding of CENP-N to the CATD, CENP-N also interacts with the H4 N-terminal tail[19]. Intriguingly, in the cryo-EM structures of the CENP-A nucleosome complexed with CENP-N, the orientation of the H4 N-terminal tail is similar to that in the crystal structure of the H3.1[CATD] nucleosome[18,19]. Therefore, the outward H4 N-terminal tail conformation, which enhances the preferential H4K20 monomethylation in the CENP-A nucleosome, may be prerequisite for the CENP-N binding.

The vast majority of the H4K20 residues in cells are methylated, including dimethylation and trimethylation, suggesting that the H4K20 residues in the H3 nucleosomes are monomethylated before the dimethylation and trimethylation events occur[53,54]. In fact, the monomethylated H4K20 residue is an excellent substrate for the Suv4-20 methyltransferase, and the H4K20 monomethylation is easily converted to H4K20 dimethylation or

trimethylation[55]. However, we did not detect the accumulation of dimethylated or trimethylated H4K20 in centromeres[28]. We propose that H4K20 monomethylation is preferential for the constitutive centromere-associated network (CCAN) assembly[56,57], and the H4 N-terminal tail in the CENP-A nucleosome associated with CCAN might prevent the access of the Suv4-20 methyltransferase, which maintains a higher level of H4K20 monomethylation in centromeres. We propose that the efficient H4K20 monomethylation in the CENP-A nucleosome promotes the preferential assembly for CCAN. Further studies will address this issue.

In the present study, we demonstrated that a few amino acid residues alter the local conformation of the nucleosomal histones. Importantly, this local structural change of the nucleosomal histones (e.g., the H4 N-terminal tail in the CENP-A nucleosome) causes a dramatic alteration of the histone modification profile of nucleosomes, possibly in collaboration with additional nucleosome binding components. This important evidence revealed that a histone variant modulates the nucleosomal histone modification, and our observation provides mechanistic insights for histone modification. We believe that certain histone modifications in the globular histone-fold domain may alter the local histone structure, thus inducing other histone modifications in the nucleosome. Although further analyses are necessary, we propose the concept that local structural changes in nucleosomes confer various functions by different histone modifications.

## Methods

**Histones and histone mutants.** For H3.1[CATD], the DNA region encoding amino acid residues 75–112 of human histone H3.1 was replaced by the DNA sequence encoding amino acid residues 75–114 of human CENP-A. For CENP-A[QD], the codons for the human CENP-A Val76 and Lys77 residues were replaced by those for Gln and Asp, respectively. For H3.1[CATD(V76Q, K77D)], the codons for the H3.1[CATD] Val76 and Lys77 residues were replaced by those for Gln and Asp, respectively. For the selenomethionine (Se-Met)-substituted H2A, the codons for H2A Leu51, Leu58, and Leu93 were replaced by the methionine codon (H2A [L51M, L58M, L93M]). The DNA primers used for the cloning were listed in Supplementary table 2.

**Purification of histones.** All histones were bacterially expressed as His$_6$-tagged recombinant proteins[12,58–60]. Human histones H2A, H2B, H3.1, H3.1[CATD], and H3.1[CATD(V76Q, K77D)] were produced in E. coli BL21(DE3). CENP-A, CENP-A[QD] was produced in E. coli BL21-CodonPlus(DE3)-RIL. Human histone H4 was produced in E. coli JM109(DE3). The Se-Met-substituted H2A [L51M, L58M, L93M] and Se-Met-substituted H2B were produced in E. coli B834(DE3). Expressed histones were corrected from the inclusion body and purified by Ni affinity chromatography using Ni-NTA beads (QIAGEN) under denaturing condition. His$_6$-tags of these purified histones were removed by treatment with thrombin protease (Wako or GE Healthcare) under non-denaturing condition. After thrombin protease treatment, histones were further purified by ion exchange chromatography using MonoS column (GE Healthcare) under denaturing condition. Purified histones were dialyzed against water containing 2 mM 2-mercaptoethanol and lyophilized. Lyophilized histone powders were stored at 4 °C.

**Preparation of nucleosomes.** For crystallization, the H2A [L51M, L58M, L93M] (Se-Met-substituted)-H2B (Se-Met-substituted)-H3.1[CATD]-H4 octamer and the H2A-H2B-H3.1[CATD(V76Q, K77D)]-H4 octamer were reconstituted with freeze-dried histones. The reconstituted histone complexes were loaded onto a HiLoad 16/600 Superdex 200 pg (GE Healthcare) and purified by gel filtration chromatography[43]. The nucleosomes were reconstituted by the salt dialysis method, with the histone octamer and a 146 base-pair palindromic human α-satellite DNA[61].

For the in vitro assays, the nucleosomes containing H3.1, CENP-A, H3.1[CATD], or CENP-A[QD] were prepared. The H2A-H2B-H3.1-H4 octamer, the H2A-H2B-CENP-A-H4 octamer, the H2A-H2B- H3.1[CATD]-H4 octamer, and the H2A-H2B-CENP-A[QD]-H4 octamer were reconstituted with freeze-dried histones. The reconstituted histone complexes were purified by gel filtration chromatography on a HiLoad 16/600 Superdex 200 pg (GE Healthcare) column. The nucleosomes were then reconstituted by the salt dialysis method, with histone octamers and the 156 base-pair palindromic DNA containing three-base 5′ overhangs on both strands. The 156 base-pair DNA sequence is: 5′-<u>AAT</u>CCAGGATCGACAATCCCGGTGC CGAGGCCGCTCAATTGGTCGTAGA<u>CAG</u>CTCTAGCACCGCTTAAACGCAC GTACGAATTCGTACGTGCGTTTAAGCGGTGCTAGAGCTGTCTACGACCA ATTGAGCGGCCTCGGCACCGGGATTGTCGATCCTGG-3′ (DNA overhang is underlined). The DNA sequence was based on the Widom 601 sequence[62]. The DNA primers used for the cloning were listed in Supplementary table 2.

The reconstituted nucleosomes were purified by native polyacrylamide gel electrophoresis using a Prep Cell model 491 apparatus (Bio-Rad)[43,59].

**Crystallization and structural determination.** The purified nucleosome containing Se-Met-substituted H2A [L51M, L58M, L93M], Se-Met-substituted H2B, and H3.1[CATD] and the nucleosome containing H3.1[CATD(V76Q, K77D)] were each dialyzed against 20 mM potassium cacodylate buffer (pH 6.0) containing 1 mM EDTA. Crystals of the nucleosomes were obtained by the hanging drop method. For the H3.1[CATD] nucleosome, the dialyzed nucleosome (1 μl) was mixed with 1 μl of 20 mM potassium cacodylate buffer (pH 6.0), containing 50 mM KCl and 85–95 mM MnCl$_2$, and was equilibrated with a reservoir solution (20 mM potassium cacodylate (pH 6.0), 40 mM KCl, and 50 mM MnCl$_2$). The nucleosome crystals were soaked in cryo-protectant buffer, containing 20 mM potassium cacodylate (pH 6.0), 40 mM KCl, 70 mM MnCl$_2$, 30% PEG400, and 5% trehalose. For the H3.1[CATD(V76Q, K77D)] nucleosome, crystals were generated and soaked in a cryo-protectant solution by the same method, using the following droplet solution (20 mM potassium cacodylate buffer (pH 6.0), 50 mM KCl, and 0.11 M MnCl$_2$), reservoir solution (20 mM potassium cacodylate (pH 6.0), 40 mM KCl, and 65 mM MnCl$_2$), and cryo-protectant solution (20 mM potassium cacodylate (pH 6.0), 29 mM KCl, 58 mM MnCl$_2$, 30% PEG400, and 5% trehalose). The crystals were flash-cooled in a stream of N$_2$ gas (100 K). The diffraction data were collected at the BL41XU station of SPring-8, Harima, Japan. For the H3.1[CATD] nucleosome, the diffraction data were integrated and scaled with the HKL2000 program and the CCP4 program suite[63,64]. For the H3.1[CATD(V76Q, K77D)] nucleosome, the diffraction data were subjected to integration, scaling, and FreeR flag generation with the XDS program suite[65,66]. The resolution limit was determined with the Aimless software[67]. The crystal structures were determined by molecular replacement with the Phaser program, using the human canonical nucleosome structures (PDB ID: 3AFA for the H3.1[CATD] nucleosome, and modified 2CV5, in which the atomic coordinates of histone H3 were removed, for the H3.1[CATD(V76Q, K77D)] nucleosome) as the search models[42,43,68]. The structural models were refined using the Phenix program suite and the Coot program[69,70]. For the H3.1[CATD] nucleosome, 98.65%, 1.35%, and 0% of the amino acids were assigned in the Ramachandran favored, allowed, and outlier regions, respectively. For the H3.1[CATD(V76Q, K77D)] nucleosome, 97.56%, 2.44%, and 0% of the amino acids were assigned in the Ramachandran favored, allowed, and outlier regions, respectively. To calculate an unbiased $F_o–F_c$ omit map, the atomic coordinates for the H4 1–25 residues were deleted from the H3.1[CATD] nucleosome structure. With this model structure, the $F_o–F_c$ omit map was calculated by the Phenix program. Structural graphics were displayed using the PyMOL program (http://pymol.org) or Chimera program[71].

**PR-Set7 purification.** The DNA fragment encoding human PR-Set7 (KMT5A Isoform 2; Uniprot ID: Q9NQR1-2) was inserted into the modified pET15b vector, which harbors the His$_6$ tag and the PreScission protease recognition sequence. His$_6$-tagged PR-Set7 was produced in *E. coli* BL21(DE3), with induction by isopropyl β-D-1-thiogalactopyranoside. After the cells were disrupted, the His$_6$-tagged PR-Set7 was recovered from the soluble fraction and purified by Ni-NTA affinity chromatography. Fractions containing His$_6$-tagged PR-Set7 were collected, and the protein was further purified by HiLoad 16/600 Superdex 200 pg (GE Healthcare) gel filtration chromatography in 50 mM Tris-HCl (pH 7.5) buffer, containing 100 mM NaCl, 10% glycerol, and 2 mM 2-mercaptoethanol. The purified His$_6$-tagged PR-Set7 was stored at -80 °C.

**In vitro H4K20 monomethylation assay.** The purified nucleosome (0.48 μM), containing H3.1, CENP-A, H3.1[CATD], or CENP-A[QD], was incubated with His$_6$-tagged PR-Set7 (0.15 or 0.30 μM) in 5.0 μl of reaction solution, containing 10 mM HEPES-KOH (pH 7.8), 8 mM Tris-HCl (pH 7.5), 20 mM KCl, 40 μM EDTA, 80 μM of *S*-adenosylmethionine, 50 mM NaCl, 0.5 mM DTT, and 9% glycerol, at 25 °C for 1, 2, 4, and 8 min. The methyltransferase reaction was stopped by adding 5 μl of a 4% SDS solution containing 0.10 M Tris-HCl (pH 6.8), 20% glycerol, and 0.2% bromophenol blue, and the samples were boiled at 95 °C for 15 min. For the western blotting analysis, 5 μl portions of the samples were analyzed by 18% SDS-PAGE, using a gel prepared with WIDE RANGE gel preparation buffer for PAGE (Nacalai Tesque), at 200 V for 130–200 min. After SDS-PAGE, the proteins were blotted onto an Amersham Hybond 0.2 μm PVDF membrane (GE Healthcare), using a Trans-Blot SD Semi-Dry Transfer Cell (Bio-Rad). The membrane was blocked with phosphate buffered saline (Takara BIO INC.), containing 5% skim milk and 0.05% Tween 20, for 1–5 h at 4 °C. The membranes were washed and then incubated with a primary antibody solution containing 1 μg/ml mouse monoclonal antibody against monomethylated H4K20 (CMA421)[49] and an anti-H2B monoclonal antibody (53H3: Cell Signaling), diluted 10,000-fold with Can Get Signal solution 1 (TOYOBO), at 4 °C overnight. After the primary antibody reaction, the membrane was washed with phosphate buffered saline containing 0.05% Tween 20 (PBS-T). The secondary antibody was Amersham ECL mouse IgG, HRP-linked F(ab')$_2$ fragment from sheep (NA9310: GE Healthcare), diluted 10,000-fold with Can Get Signal solution 2 (TOYOBO), and it was incubated with the membrane for 1–3 h at 4 °C. The membrane was washed with PBS-T, and then treated with Amersham ECL Prime (GE Healthcare). Chemiluminescence was detected with an Amersham Imager 680 (GE Healthcare). The uncropped images of all blots are presented in the Supplementary Figure 8.

**Nucleosome proteinase accessibility assay.** The purified nucleosome (0.38 μM), containing H3.1 or CENP-A, was incubated with trypsin (2.5, 5.0, 7.5, 10 ng, purchased from Sigma-Aldrich) in 10 μl of reaction solution, containing 8 mM Tris-HCl (pH 7.5), 0.5 mM MgCl$_2$, and 1 mM DTT, at 25 °C for 1 min. The reaction was stopped by adding 10 μl of a 4% SDS solution, containing 0.10 M Tris-HCl (pH 6.8), 20% glycerol, and 0.2% bromophenol blue, and the samples were boiled at 95 °C for 15 min. For the western blotting, 6.0 μl portions of the samples were analyzed by 18% SDS-PAGE at 200 V for 100 min. After SDS-PAGE, the western blotting analyses were performed by the method described above, using an anti-H4 rabbit polyclonal antibody (1:1000; Abcam, ab7311) and an HRP-anti rabbit-IgG F(ab')$_2$ fragment (1:5000; GE Healthcare, NA9340) as the primary and secondary antibodies, respectively. The blocking, primary antibody reaction, and secondary antibody reaction were performed at 4 °C overnight, 4 °C for 3–12 h, and 4 °C for 1–2 h, respectively. The membrane was washed with PBS-T, and then treated with Amersham ECL Prime (GE Healthcare). Chemiluminescence was detected with an LAS4000 image analyzer (GE Healthcare). The uncropped images of all blots are presented in Supplementary information. The uncropped images of all blots are presented in Supplementary Figure 11.

**Gel shift assay with PR-Set7 and nucleosome.** The purified nucleosome (0.48 μM), containing H3.1 or CENP-A, was incubated with His$_6$-tagged PR-Set7 (0.49, 0.97, 1.5, and 1.9 μM) in 5.0 μl of reaction solution, containing 10 mM HEPES-KOH (pH 7.8), 18 mM Tris-HCl (pH 7.5), 50 mM KCl, 0.10 mM EDTA, 80 μM of *S*-adenosylmethionine, 50 mM NaCl, 0.50 mM DTT, and 15% glycerol, at 25 °C for 30 min. After the incubation, the samples were analyzed by 6% native PAGE with 0.2X TBE buffer at 150 V for 60 min. The gels were stained with SYBR-Gold and imaged with an LAS4000 image analyzer (GE Healthcare).

**Cell culture.** Chicken DT40 cells were cultured in DMEM medium (Nacalai Tesque), supplemented with 10% fetal bovine serum (FBS; Sigma), 1% chicken serum (Gibco), 10 μM 2-mercaptoethanol, and Penicillin/Streptomycin (final: 100 unit/ml and 100 μg/ml, respectively) (Thermo Fisher), at 38.5 °C with 5% CO$_2$[72]. *Drosophila* S2 cells (Gibco) were cultured at 28 °C in Schneider's *Drosophila* medium (Gibco), supplemented with 10% FBS, and Penicillin/Streptomycin (final: 50 unit/ml, 50 μg/ml, respectively).

**Plasmid constructions for ChIP-seq.** To express GFP-fused chicken CENP-A, the cDNA encoding full-length chicken CENP-A was cloned into pEGFP-C2 (Clontech) (ggCENP-A_pEGFP-C2). The cDNA fragment of the chicken CENP-A[QD] mutant, which has two amino acid substitutions (L67Q and L68D), was synthesized by Gene Synthesis (FASMAC), and was cloned into pEGFP-C2 to yield the GFP-CENP-A[QD]_pEGFP-C2 expression vector (ggCENP-A[QD]_pEGFP-C2).

**Generation of cell lines.** To generate DT40 cells expressing GFP-CENP-A or GFP-CENP-A[QD], ggCENP-A_pEGFP-C2 or ggCENP-A[QD]_pEGFP-C2 plasmids containing a Blasticidin resistance gene were transfected into CENP-A conditional knockout cells [CENP-A (–/Flox), Mer-Cre-Mer][22] with a Gene Pulser II electroporator (Bio-Rad). Cells transfected with the ggCENP-A_pEGFP-C2 plasmid were selected in medium containing 2 mg/ml G418 (Santa Cruz Biotechnology), and cells transfected with the ggCENP-A[QD]_pEGFP-C2 plasmid were selected in medium containing 25 μg/ml Blasticidin S hydrochloride (Wako). After 10 days of selection, the drug resistant and GFP-positive clones were isolated. To knockout the endogenous CENP-A gene in the isolated clones, 100 nM 4-hydroxytamoxifen (OHT, Sigma) was added to the culture medium to activate the Mer-fused Cre-recombinase (Mer-Cre-Mer), and then the 5′ portion of the CENP-A gene [from exon 1 to 3 (CENP-A 1–87)] flanked by the LoxP sequences was excised from the genome. After the OHT treatment, we further isolated the monoclonal lines by limited dilution in 96-well plates, and verified the depletion of endogenous CENP-A in the isolated clones by a southern blot analysis. CENP-A disruption was also confirmed by an immunoblot analysis.

**Immunoblot analyses of DT40 whole-cell extracts.** Whole-cell extracts were prepared from the DT40 cell lines, and were separated by SDS-PAGE. The gels were transferred onto PVDF membranes (FluoroTrans, PALL). The membranes were blocked with 5% skim milk (BD), and then probed with the following antibodies: rabbit anti-GFP (1:5000; MBL), rabbit anti-chicken CENP-A (1:5000)[73], mouse anti-H4K20 monomethylation (1:5000; #CMA421)[28,49], or Rat anti-pan H3 (1:5000; #140-1G1)[74] as the primary antibody. HRP-conjugated anti-rabbit IgG (1:15,000; Jackson ImmunoResearch; 111-035-144) and HRP-conjugated anti-mouse IgG (1:15,000; Jackson ImmunoResearch; 115-035-003) were used as secondary antibodies. Signals were developed using ECL Prime (GE Healthcare), and were detected by a ChemiDoc Touch imaging system (Bio-Rad). The uncropped images of all blots are presented in the Supplementary Figures 12 and 13.

**Spike-in ChIP-seq.** To normalize the ChIP-seq data for quantitative comparisons of H4K20me1 or CENP-A, we employed the spike-in normalization method using *Drosophila* chromatin as the references[50,51]. Nuclei were prepared from the CENP-A-deficient DT40 cells ($1 \times 10^8$) expressing GFP-CENP-A or GFP-CENP-A[QD]. Nuclei from *Drosophila* S2 cells (Gibco, $7.7 \times 10^7$) were also prepared. Nuclei samples were digested with MNase (final 2000 Gel U/ml; NEB) in 1 ml of buffer A

[15 mM HEPES-NaOH (pH 7.4), 15 mM NaCl, 60 mM KCl, 0.34 M sucrose, 0.5 mM spermine, 0.15 mM spermidine, 1 mM dithiothreitol, 1 mM $CaCl_2$, and cOmplete protease inhibitor cocktail (-EDTA); Roche] for 1 h at 37 °C, followed by the extraction of the mono-nucleosome fraction in buffer containing 0.5 M NaCl. The fixed amount of the *Drosophila* S2 mono-nucleosome fraction was added to the DT40 mono-nucleosome fraction. For the chromatin IP (ChIP) with anti-H4K20me1, the DT40 mono-nucleosome (21.8 μg of DNA) was mixed with the *Drosophila* S2 mono-nucleosome (218 ng of DNA) at a ratio of 100:1. For the ChIP with anti-GFP, the DT40 mono-nucleosome (65.4 μg of DNA) was mixed with the *Drosophila* S2 mono-nucleosome (218 ng of DNA) at a 300:1 ratio. The mixed mono-nucleosomes were incubated for 2 h at 4 °C with 45 μl of Dynabead-conjugated protein G (Invitrogen), which was pre-incubated with 5 μg of mouse anti-H4K20me1 and 2.5 μg of rabbit anti-H2Av (Spike-in antibody; Active Motif) in buffer B [20 mM Tris-HCl (pH 8.0), 5 mM EDTA, 500 mM NaCl, 0.2% Tween 20, and cOmplete protease inhibitor cocktail (-EDTA)]. The samples were also incubated for 2 h at 4 °C with 45 μl of Dynabead-conjugated protein G, which was pre-incubated with 5 μl of rabbit anti-GFP and 2.5 μg of rabbit anti-H2Av in buffer B. The beads were washed four times with buffer B, and the bound DNA was purified with a MinElute PCR purification kit (Qiagen). ChIP-seq libraries were constructed using an NEBNext Ultra II DNA library prep Kit for Illumina (NEB), according to the manufacturer's protocol. Thirty nanograms of purified DNA was end-repaired and ligated to the adaptor for Illumina, and then subjected to 8 PCR cycles with the universal PCR primers with the index sequences. ChIP-seq libraries were sequenced using an Illumina HiSeq 2500 (100 bp single end sequencing). The sequence data were mapped to the Chicken Genome database (NCBI, Gallus_gallus-4.0), as well as to the *Drosophila* genome database (NCBI, *Drosophila*-dm6), with the Burrows-Wheeler Aligner version 0.6.2 mapping tool[75]. Correction factors for ChIP-seq with anti-H4K20me1 or with anti-GFP were generated independently, using the numbers of sequence-reads mapped on the *Drosophila* genome. The numbers of sequence-reads mapped on the Chicken genome were then normalized, using the correction factors.

**Reporting summary**. Further information on experimental design is available in the Nature Research Reporting Summary linked to this article.

## Data availability

The coordinates and structure factors of the H3.1$^{CATD}$ nucleosome and the H3.1$^{CATD(V76Q, K77D)}$ nucleosome have been deposited in the Protein Data Bank, under the accession codes 5Z23 and 5ZBX, respectively. Spike-in ChIP-seq data with anti-H4K20me1 or anti-GFP antibodies were submitted to the DDBJ Sequence Read Archive. Accession codes and FTP download links are as follows: SAMD00132315 (gfpCA1-1_input) via ftp://ftp-trace.ncbi.nlm.nih.gov/sra/sra-instant/reads/ByRun/sra/DRR/DRR144/DRR144862 SAMD00132316 (S2_input) via ftp://ftp-trace.ncbi.nlm.nih.gov/sra/sra-instant/reads/ByRun/sra/DRR/DRR144/DRR144863 SAMD00132317 (gfpCA1-1_IgG_SpI) via ftp://ftp-trace.ncbi.nlm.nih.gov/sra/sra-instant/reads/ByRun/sra/DRR/DRR144/DRR144864 SAMD00132318 (gfpCA-QD2-5_IgG_SpI) via ftp://ftp-trace.ncbi.nlm.nih.gov/sra/sra-instant/reads/ByRun/sra/DRR/DRR144/DRR144865 SAMD00132319 (gfpCA-QD3-1_IgG_SpI) via ftp://ftp-trace.ncbi.nlm.nih.gov/sra/sra-instant/reads/ByRun/sra/DRR/DRR144/DRR144866 SAMD00132320 (gfpCA1-1_GFP_SpI) via ftp://ftp-trace.ncbi.nlm.nih.gov/sra/sra-instant/reads/ByRun/sra/DRR/DRR144/DRR144867 SAMD00132321 (gfpCA-QD2-5_GFP_SpI) via ftp://ftp-trace.ncbi.nlm.nih.gov/sra/sra-instant/reads/ByRun/sra/DRR/DRR144/DRR144868 SAMD00132322 (gfpCA-QD3-1_GFP_SpI) via ftp://ftp-trace.ncbi.nlm.nih.gov/sra/sra-instant/reads/ByRun/sra/DRR/DRR144/DRR144869 SAMD00132323 (gfpCA1-1_H4K20me1_SpI) via ftp://ftp-trace.ncbi.nlm.nih.gov/sra/sra-instant/reads/ByRun/sra/DRR/DRR144/DRR144870 SAMD00132324 (gfpCA-QD2-5_H4K20me1_SpI) via ftp://ftp-trace.ncbi.nlm.nih.gov/sra/sra-instant/reads/ByRun/sra/DRR/DRR144/DRR144871 SAMD00132325 (gfpCA-QD3-1_H4K20me1_SpI) via ftp://ftp-trace.ncbi.nlm.nih.gov/sra/sra-instant/reads/ByRun/sra/DRR/DRR144/DRR144872. A reporting summary for this Article is available as a Supplementary Information file. All other data supporting the findings of this study are available from the corresponding author on reasonable request.

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

## Acknowledgements

We are grateful to Ms. Y. Iikura (The University of Tokyo) and Mr. T. Shiga (Waseda University) for their assistance. We thank the beamline scientists for their assistance with data collection at the BL41XU beamline of SPring-8. The synchrotron radiation experiments were performed with the approval of the Japan Synchrotron Radiation Research Institute (JASRI) [proposal no. 2010A1206]. This work was supported in part by the JSPS KAKENHI Grant Numbers JP17K15043 [to Y.A.], JP18H05534 [to H. Kurumizaka], JP25116002 [to H.Kurumizaka and T.H.], JP17H01408 [to H.Kurumizaka], JP16K14785 [to H.Tachiwana], JP17H05013 [to H.Tachiwana], JP17K07501 [to T.H.], JP15H05972 [to T.F.], JP17H06167 [to T.F.], JP18H05527 [to H.Kimura], and JP25116005 [to H.Kimura]. This work was partly supported by JST CREST Grant Number JPMJCR16G1 [to H.Kurumizaka and H.Kimura] and by the Platform Project for Supporting Drug Discovery and Life Science Research (BINDS) from AMED under Grant Number JP18am0101076 [to H.Kurumizaka].

## Author contributions

Y.A., H.Tachiwana, and H.Takagi reconstituted the nucleosomes with the H3.1$^{CATD}$ and the histone mutants, collected X-ray diffraction data, and determined the crystal structures. H.Takagi and H.Kimura contributed to western blotting analyses. T.H. and T.F. contributed to genomics analyses. H.Kurumizaka conceived, designed, and supervised all of the work, and wrote the paper. All of the authors discussed the results and commented on the manuscript.

## Additional information

**Competing interests:** The authors declare no competing interests.

