## [Peer Review File · Nature Communications]

Reviewers' comments:

Reviewer #1 (Remarks to the Author):

Arimura et al. describe a molecular mechanism by which replacement of histone H3 by the centromere-specific CENP-A molecule facilitates methylation of H3K20, found to exist in centromeric nucleosomes. The basis of the mechanism is that the replacement of two H3-specific residues, Gln76 and Asp77 by a valine and lysine in the CENP-A specific CATD motif, release the tail of histone H4 to the solvent, where it is presumably accessible to the requisite methylase activity.

If correct, this would provide an interesting mechanism for control of CENP-A PTMs, and the authors indeed show biochemically and in vivo that mutation of CENP-A to contain the Gln76/Asp77 pair inhibit H3K20 methylation to some extent. However, I have some serious reservations about the interpretation of the structural data that underpins the proposed mechanism. As the manuscript stands, I do not believe that the presented data support the principal hypothesis, and so cannot support publication for the following reasons:

1. The basis for the sequestering of the H4 tail in H3 nucleosomes is suggested to predominantly be an interaction between the carboxyl side-chain of H3 Asp77 and the main-chain carbonyl of Leu22. Analysis of some available H3 nucleosome structures (5AV6, 2CV5, 3AFA, 3LJA, and see point 2 below) shows that it is far from clear whether this interaction really exists. From figure 2a, it appears the interaction is mediated by a coordinated manganese ion, bridging to the Leu22 carboxyl by two water molecules. No bond distances are shown on the figure, making it difficult to assess the likelihood of these interactions. However, looking at the H3 nucleosome structure PDB ID:2CV5, it seems that while the Mn²⁺ ion can coordinate with the Asp77 side-chain, it is > 4Å away from the Leu22 carbonyl – well outside the usual coordination sphere (~2.2Å for manganese). The bridging water molecule shown in the figure does not seem to exist in any other structure. Furthermore, the metal coordination and water placement vary considerably among deposited

structures, and even where the Mn²⁺ ion is present, it appears to be stabilised by symmetry-related molecules – i.e. its binding may well be a function of crystal packing.

Aside from these points, it is striking that the H4 tail is only modelled and presumably partially ordered in one of the two H4 molecules in the nucleosome. In the structures listed above, the equivalent region of the tail including Leu22 is either missing (2CV5, 3AFA, 3LJA), or running in the opposite direction (5AV6) in the dyad-related H4. If the tail were indeed held in place by a stable interaction with H3 as the authors propose, one would expect to see a consistent conformation for this region both within and between nucleosome structures. This clearly is not the case.

2. The H3.1 nucleosome structure used to generate figures 1a, 1d and 2f is given the PDB accession code 5Y0C and described as “submitted elsewhere”. Checking the PDB reveals that this structure is deposited and on hold (i.e. unreleased). Without access to these coordinates, the reviewer has no way of assessing the accuracy of the figures or making independent checks of the proposed interactions in this figure. The authors may wish to reconsider referring to unpublished and inaccessible data to support a key argument in their paper.

3. It is not clear from the manuscript why the authors chose to solve a chimeric H3^{CATD} nucleosome structure rather than a genuine CENP-A nucleosome. Presumably it was motivated by the fact that the N-terminal H4 tail in the extent CENP-A crystal structure is not visible, but why the tail should be visible in the H3^{CATD} used in this study and not CENP-A has not been rationalised. Again, the lack of structural consistency makes it a little hard to accept that there are clearly defined conformations for the H4 N-terminus, which undermines the mechanisms proposed in this work.

4. Extended data figure 1. The electron density map for the H4 tail in the H3^{CATD} structure is a refined 2Fo-Fc map. To provide an accurate assessment of the density quality for the structure in this region (and support the modelled tail conformation), the authors should show an unbiased Fo-Fc omit map at stated contour level (preferably 3 σ).

Reviewer #2 (Remarks to the Author):

Manuscript by Kurumizaka and colleagues describes two structures of H3.1-CENPA chimeric nucleosome. Specifically, CATD of CENP-A, the domain responsible for centromere targeting replaces part of H3.1 in one of their structures. Based on this structure and biochemical and functional analyses they propose a mechanism by which two residues in CATD domain V76 and K77 cancel local interactions with histone H4 thereby changing the accessibility of H4K20 and facilitating its methylation, important feature of CENP-A nucleosomes in vivo. The other structure contains the same chimeric nucleosome with these two residues mutated which is proposed to revert the effect.

I think this is an interesting hypothesis but it needs further validation to be convincing. While they show different location of H4 tail in H3.1 vs chimera – depending which face of the nucleosome is looked at -both locations of H4 can be found in previously determined structures of nucleosomes

that have nothing to do with CENP-A (examples of pDBs: 1KX5, 2NQB, 5AVB, 1S32). I am wondering if the location of the tail is a function of crystal packing and not of a particular feature of this nucleosome. It is essential that the authors show both sides of their crystallized nucleosomes and comment on independence of this finding from crystal packing to support their data. They should compare their structures with both sides of the previous published structures.

Additionally, it would be good if they could validate their mechanistic hypothesis by performing some solution method that could directly show the altered accessibility of H4.

While they do show methyltransferase data to support their structures it is difficult to reach conclusions based on a single point enzymatic assay that does not show the input enzyme and also does not show the titration. They should either do a proper kinetic analysis or at least show all necessary controls.

The in vivo data are difficult to interpret. Since there is less GFP signal in ChIP-seq of the mutant (QD) than that of the WT (CENP-A) protein it is not easy to evaluate the impact of methylation here. Are recruitment and incorporation impacted in the mutant? If so one would naturally expect lower levels of methylation but not necessarily as a function of H4 accessibility. The experiment should be repeated (or normalized?) in a manner where GFP expression for both the wild type and mutant is the same.

I think that the authors should address the comments above in order to support their hypothesis.

Additionally, I have small points to address:

- Can the authors include the difference map in the region of interest (again both sides of the nucleosome)? I would recommend to show the density contoured also at lower sigma (than 1 at which some side-chain density is missing judging by their figures), include the *CC value for the residues and/or include a simulated annealing omit map for the region.
- I would also include a table with the B factors of the different regions including the H4 tail, and other comparing the B factor of the Mn ions/structural waters and the surrounding residues (V76K77).

Reviewer #3 (Remarks to the Author):

In this study, Arimura et al. use X-ray crystal structures of nucleosomes containing WT H3 (H3.1), CENP-A and H3 with the CENP-A centromere targeting domain (CATD) swapped into H3.1 nucleosomes to demonstrate that the H4 tail domain adopts distinct conformations in the crystal structures of these nucleosomes, dependent on two residues in H3.1 (Q76 and D77) that interact with the H4 tail, and constrain its trajectory, while the corresponding residues in the H3 CATD protein (V76 and K77) do not interact with the tail, allowing a more unconstrained structure. The authors hypothesize that the different H4 tail structures explains the localization of monomethylated H4 K20 in CENP-A nucleosomes, suggesting that the residue in H3.1 nucleosomes would be less available for modification. Indeed in vitro methylation assays with the monomethylase PR-Set7 show a faster methylation of H4 K20 in CENP-A or H3.1 CATD nucleosomes, dependent on V76 and K77. Moreover, ectopic expression of GFP-tagged chicken CENP-A or the CENP-A mutant V67Q/L68D (corresponding to the same residues in the chicken histones) in chicken DT40 cells showed less localization to CEN DNA when ChIP'ed with antibody against monomethylated H4K20 but not with GFP, suggesting monomethylation of CENP-A is dependent on the two CATD residues identified in the structural studies. The data shown is of high quality and compelling.

The authors conclude that the structural state of the H4 tail, which is defined by interaction with H3.1 (or lack thereof (CENP-A)) in the nucleosome ultimately regulate the activity of the PR-Set7 monomethylase. While an interesting conjecture, I feel that there are some significant questions regarding this conclusion.

First, could the two residues in question alter the binding of PR-Set7 to the nucleosome? One might imagine that a rather small change in substrate binding free energy, equal to about a single H-bond, could account for the apparent 6-fold change in rate. This would be completely independent of the state of the H4 tail, which is assumed to be quite dynamic. If the enzyme interacts with the CATD domain, this would indeed be a possibility. This is not addressed in the paper, either experimentally or in the discussion.

Second, a major problem with the main conclusion is the fact that the vast majority of H4 in most cells is methylated at K20. Indeed, estimates run from >98% (Pesavento, 2008 MCB doi: 10.1128/MCB.01517-07) to ~85 % in a latter study (Huang, 2015 Chem Rev doi: 10.1021/cr500491u). This, coupled with the fact that PR-Set7 is believed to be the only monomethylase in the cell, and, moreover, the fact that PR-Set7 monomethylation is required for subsequent higher-order methylation events at K20 (K20me2, K20me3) by other enzymes indicates that for the vast majority of nucleosomes (i.e. major H3 nucleosomes) access to the site of modification is not limiting. This makes the proposed model quite untenable. It seems to me that the authors must develop their model within this context, or provide contrary argument. For example, one possibility, not

considered in the MS, is that the CATD domain directly inhibits the installation of additional methyl at K20 groups by Suv4-20.

Reviewers' comments:

Reviewer #1 (Remarks to the Author):

General comment)

Arimura et al. describe a molecular mechanism by which replacement of histone H3 by the centromere-specific CENP-A molecule facilitates methylation of H3K20, found to exist in centromeric nucleosomes. The basis of the mechanism is that the replacement of two H3-specific residues, Gln76 and Asp77 by a valine and lysine in the CENP-A specific CATD motif, release the tail of histone H4 to the solvent, where it is presumably accessible to the requisite methylase activity.

If correct, this would provide an interesting mechanism for control of CENP-A PTMs, and the authors indeed show biochemically and in vivo that mutation of CENP-A to contain the Gln76/Asp77 pair inhibit H3K20 methylation to some extent. However, I have some serious reservations about the interpretation of the structural data that underpins the proposed mechanism. As the manuscript stands, I do not believe that the presented data support the principal hypothesis, and so cannot support publication for the following reasons:

Comment 1-1)

The basis for the sequestering of the H4 tail in H3 nucleosomes is suggested to predominantly be an interaction between the carboxyl side-chain of H3 Asp77 and the main-chain carbonyl of Leu22. Analysis of some available H3 nucleosome structures (5AV6, 2CV5, 3AFA, 3LJA, and see point 2 below) shows that it is far from clear whether this interaction really exists. From figure 2a, it appears the interaction is mediated by a coordinated manganese ion, bridging to the Leu22 carboxyl by two water molecules. No bond distances are shown on the figure, making it difficult to assess the likelihood of these interactions.

Reply)

Thank you very much for this suggestion. To clearly show the interaction between the carboxyl side-chain of H3 Asp77 and the main-chain carbonyl of Leu22, we measured and presented the bond distances in the new Extended Data Fig. 6 in the revised manuscript. The distances between the Mn²⁺ ion and the Asp77 side-chain or water molecule are 2.1 angstroms and 2.1 angstroms, respectively. The distance between the water molecule and the Leu22 main chain carbonyl is 3.4 angstroms. We hope that these additional data will convince the readers of the interaction.

Comment 1-2)

However, looking at the H3 nucleosome structure PDB ID:2CV5, it seems that while the Mn²⁺ ion can coordinate with the Asp77 side-chain, it is > 4Å away from the Leu22 carbonyl – well outside the usual coordination sphere (~2.2Å for manganese). The bridging water molecule shown in the figure does not seem to exist in any other structure. Furthermore, the metal coordination and water placement vary considerably among deposited structures, and even where the Mn²⁺ ion is present, it appears to be stabilised by symmetry-related molecules – i.e. its binding may well be a function of crystal packing.

Reply)

In the 2CV5 structure, the Mn²⁺ ion and the coordinating water molecules were not placed, probably due to its low resolution. In contrast, in the 1KX5 structure, showing the high-resolution nucleosome at 1.9 angstroms, the Mn²⁺ ion and its coordinating water molecules are clearly placed. In this structure, the distances between the Mn²⁺ ion and the Asp77 side-chain or water molecule are 2.2 angstroms and 2.3 angstroms, respectively. These distances are quite consistent with our current high-resolution nucleosome structure, shown in new Extended Data Fig. 6 (PDB ID=5Y0C).

Comment 1-3)

Aside from these points, it is striking that the H4 tail is only modelled and presumably partially ordered in one of the two H4 molecules in the nucleosome. In the structures listed above, the equivalent region of the H4 tail including Leu22 is either missing (2CV5, 3AFA, 3LJA), or running in the opposite direction (5AV6) in the dyad-related H4. If the tail were indeed held in place by a stable interaction with H3 as the authors propose, one would expect to see a consistent conformation for this region both within and between nucleosome structures. This clearly is not the case.

Reply)

Thank you very much for this important and insightful comment. As this reviewer pointed out, we noticed that one of the two H4 N-terminal tails forms a similar conformation to that in the H3.1^{CATD} nucleosome in several cases (Extended Data Fig. 5). In addition, the current Cryo-EM analyses of the H3 nucleosome revealed that the H4 N-tail conformation was similar to that of the H3.1^{CATD} nucleosome (Extended Data Fig. 4), although the population with this conformation was minor. These findings suggest that the H4 tail conformation in the H3.1^{CATD} nucleosome does not depend on crystal packing, and therefore we named these H4 N-terminal tail conformations as the inward H4-N and outward H4-N conformations.

We would like to emphasize that while the canonical H3 nucleosome forms both the inward H4-N and outward H4-N conformations, the H3.1^{CATD} nucleosome forms only the outward H4-N conformation. Since the inward H4-N conformation requires the Gln76 and Asp77 residues in H3 and these residues are absent in CENP-A, we propose that the CENP-A nucleosome does not form the inward H4-N conformation. Based on these new findings, we now explain why the H4K20 methylation efficiently occurs in the CENP-A nucleosome, although the H4K20 residue can be methylated in both the H3 and CENP-A nucleosomes. These new findings are described on p.5, ll.3-18, and are discussed in the section “The CATD V76 and K77 residues abrogate the local H3-H4 interaction” and the second paragraph of the Discussion in the revised version.

Comment 2)

The H3.1 nucleosome structure used to generate figures 1a, 1d and 2f is given the PDB accession code 5Y0C and described as “submitted elsewhere”. Checking the PDB reveals that this structure is deposited and on hold (i.e. unreleased). Without access to these coordinates, the reviewer has no way of assessing the accuracy of the figures or making independent checks of the proposed interactions in this figure. The authors may wish to reconsider referring to unpublished and inaccessible data to support a key argument in their paper.

Reply)

We apologize for this inconvenience. We have now released the high-resolution nucleosome structure used in the present study, with PDB ID: 5Y0C. We hope that this reviewer can evaluate the structure.

Comment 3)

It is not clear from the manuscript why the authors chose to solve a chimeric H3^{CATD} nucleosome structure rather than a genuine CENP-A nucleosome. Presumably it was motivated by the fact that the N-terminal H4 tail in the extant CENP-A crystal structure is not visible, but why the tail should be visible in the H3^{CATD} used in this study and not CENP-A has not been rationalised. Again, the lack of structural consistency makes it a little hard to accept that there are clearly defined conformations for the H4 N-terminus, which undermines the mechanisms proposed in this work.

Reply)

While we have determined the CENP-A nucleosome at 3.6 angstrom resolution, this resolution is not sufficient to visualize the H4 tail conformation clearly. Of course, we made various efforts to obtain a higher

resolution CENP-A (or its derivative) nucleosome structure. Unfortunately, we did not obtain better conditions in our current situation. However, we found that the nucleosome crystals containing the H3.1^{CATD}, which ensures CENP-A function *in vivo*, showed better diffraction. We then visualized the H4 tail conformation in the H3.1^{CATD} nucleosome. In the revised manuscript, we clearly mention the reason why the H3.1^{CATD} nucleosome was used in this study, by adding these explanations (p.3, ll.29-31).

Comment 4)

Extended data figure 1. The electron density map for the H4 tail in the H3^{CATD} structure is a refined 2Fo-Fc map. To provide an accurate assessment of the density quality for the structure in this region (and support the modelled tail conformation), the authors should show an unbiased Fo-Fc omit map at stated contour level (preferably 3 σ).

Reply)

We agree with this comment and present an unbiased Fo-Fc omit map at the 3 σ contour level in the new Extended Data Fig. 2. To present the map, the atomic coordinates for the H4 residues 1-25 were deleted from the H3.1^{CATD} nucleosome, and this omit map was calculated. As shown in the new Extended Data Fig. 2, the electron density map for the H4 residues 19-25 is clearly visualized in this unbiased Fo-Fc omit map.

Reviewer #2 (Remarks to the Author):

General comment)

Manuscript by Kurumizaka and colleagues describes two structures of H3.1-CENPA chimeric nucleosome. Specifically, CATD of CENP-A, the domain responsible for centromere targeting replaces part of H3.1 in one of their structures. Based on this structure and biochemical and functional analyses they propose a mechanism by which two residues in CATD domain V76 and K77 cancel local interactions with histone H4 thereby changing the accessibility of H4K20 and facilitating its methylation, important feature of CENP-A nucleosomes *in vivo*. The other structure contains the same chimeric nucleosome with these two residues mutated which is proposed to revert the effect.

Comment 1-1)

I think this is an interesting hypothesis but it needs further validation to be convincing. While they show different location of H4 tail in H3.1 vs chimera – depending which face of the nucleosome is looked at -both locations of H4 can be found in previously determined structures of nucleosomes that have nothing to do with CENP-A (examples of pids: 1KX5, 2NQB, 5AVB, 1S32).

Reply)

We investigated the PDBs for 1KX5, 2NQB, 5AVB, and 1S32.

1KX5: *Xenopus* canonical nucleosome (1.94 Å).

2NQB: *Drosophila* canonical nucleosome (2.3 Å).

5AVB: Human canonical nucleosome (2.4 Å).

1S32: *Xenopus* canonical nucleosome (2.05 Å).

This is a quite useful suggestion, and we compared the H4 N-tail conformations of their structures on both sides. Actually, only one side of the H4K20 residue can be visualized in these nucleosomes, and the opposite side of the H4K20 residue is ambiguous. However, in the 2NQB, 5AVB, and 1S32 nucleosome structures, H4L22 is observable on the ambiguous side, and the backbone configuration of the H4 N-tail on this side can be deduced. We then noticed that the H4 N-tail on this side forms a similar conformation to that in the H3^{CATD} nucleosome. This is a really important finding, and it explains why the H4K20 residue is preferentially methylated in the CENP-A nucleosome, although it is also methylated in the canonical H3 nucleosome. Although the two H4 tails can form both inward and outward H4-N conformations in the H3 nucleosome, the H3.1^{CATD} nucleosome can only form the outward H4-N conformation. We believe that this explains why the H4K20 residues of the H3.1^{CATD} nucleosome are more efficiently monomethylated than those of the H3 nucleosome. These new findings are presented in the new Extended Data Fig. 5, and are discussed in the revised manuscript.

Comment 1-2)

I am wondering if the location of the tail is a function of crystal packing and not of a particular feature of this nucleosome. It is essential that the authors show both sides of their crystallized nucleosomes and comment on independence of this finding from crystal packing to support their data. They should compare their structures with both sides of the precious published structures.

Reply)

In our H3.1^{CATD} nucleosome structure, one H4 N-tail is clearly visualized, but the other side is ambiguous. However, in the 2Fo-Fc map, the Arg23 residue on the ambiguous side of the H4 tail is detectable (Extended Data Fig. 1), and we confirmed that both H4 N-tail orientations in the H3.1^{CATD} nucleosome are similar, suggesting that our observation is not a crystal packing artifact. In addition, the current Cryo-EM analyses revealed that the H4 N-tail adopts the outward conformation in the CENP-A nucleosome in solution (Extended Data Fig. 4). These findings also support our conclusion that the H4 tail conformation in

the H3.1^{CATD} nucleosome does not depend on crystal packing. These new data are described in the Results section “Crystal structure of the nucleosome containing histone H3.1 with CATD”.

Comment 2)

Additionally, it would be good if they could validate their mechanistic hypothesis by performing some solution method that could directly show the altered accessibility of H4.

Reply)

We agree with this comment and performed the proteinase accessibility assay using the H4 N-terminal tails *in vitro*. Based on this assay, the H4 N-terminal tails of both the CENP-A and H3 nucleosomes were trypsin-accessible to similar extents (new Fig. 3e and Extended Data Fig. 9), indicating that the H4 tail accessibility itself may be similar between the H3 and CENP-A nucleosomes. In contrast, we found that the binding stability of PR-Set7 to the CENP-A nucleosome is substantially different from that to the H3 nucleosome. In fact, the interaction between PR-Set7 and the CENP-A nucleosome is less stable than that between PR-Set7 and the H3 nucleosome, which caused elevated PR-Set7 turnover. This suggested that the outward H4-N conformation of the CENP-A nucleosome may be a more efficient H4K20 monomethylation substrate for PR-Set7, as compared with the H3 nucleosome. We present these new data in the new Fig. 3f and Extended Data Fig. 9, and the results are described in the Results section “CATD stimulates H4K20 monomethylation by PR-Set7 in the nucleosome” of the revised manuscript.

Comment 3)

While they do show methyltransferase data to support their structures it is difficult to reach conclusions based on a single point enzymatic assay that does not show the input enzyme and also does not show the titration. They should either do a proper kinetic analysis or at least show all necessary controls.

Reply)

We agree with this comment and performed time course analyses of the methyltransferase assay with the nucleosomes containing H3, CENP-A, H3.1^{CATD}, and the CATD mutant. We then confirmed that these results are consistent with our conclusion. We present these new data in the new Extended Data Fig. 8.

Comment 4)

The *in vivo* data are difficult to interpret. Since there is less GFP signal in ChIP-seq of the mutant (QD) than that of the WT (CENP-A) protein it is not easy to evaluate the impact of methylation here. Are recruitment

and incorporation impacted in the mutant? If so one would naturally expect lower levels of methylation but not necessarily as a function of H4 accessibility. The experiment should be repeated (or normalized?) in a manner where GFP expression for both the wild type and mutant is the same.

Reply)

To address these comments, we performed spike-in ChIP-seq experiments, as shown in Figure 4c. In these experiments, a fixed amount of a chromatin sample from *Drosophila* was added to our experimental samples from chicken cells, and we performed ChIP-seq analyses with a target antibody (either anti-GFP or -H4K20me1) and an anti-*Drosophila* H2Av antibody. We mapped the sequence data into the chicken and *Drosophila* genome databases, and the sequence reads in the chicken genome were normalized to the read-counts mapped to the *Drosophila* genome. We used two independent CENP-A knockout chicken DT40 cell lines expressing GFP-CENP-A^{QD} (clones #2-5 and #3-1). The expression level of GFP-CENP-A^{QD} in the #2-5 clone was similar to that of GFP-CENP-A in cells expressing GFP-CENP-A, and the expression of GFP-CENP-A^{QD} in the #3-1 clone was slightly higher than that in the #2-5 clone (Extended Data Figures 10 and 14). GFP-ChIP accumulation around the centromere region in both cell lines expressing GFP-CENP-A^{QD} (#2-5 and #3-1) was slightly lower (~80% level) than that in cells expressing GFP-CENP-A, although non-centromeric CENP-A was increased in the #2-5 and #3-1 clones, probably due to the presence of excess GFP-CENP-A^{QD} (Figure 4d, e and f). The H4K20me1 levels around the centromere region were substantially lower in both the #2-5 and #3-1 cell lines (approximately 40%) than that in cells expressing GFP-CENP-A, but these levels in non-centromeric regions were constant (Figure 4d, e and f). Based on the results with the quantitative spike-in ChIP-seq analyses using two independent clones, we conclude that the H4K20me1 level in the centromere region was significantly reduced in cells expressing the CENP-A^{QD} mutant.

Minor points)

I think that the authors should address the comments above in order to support their hypothesis.

Additionally, I have small points to address:

Comment 1)

- Can the authors include the difference map in the region of interest (again both sides of the nucleosome)?

I would recommend to show the density contoured also at lower sigma (than 1 at which some side-chain density is missing judging by their figures), include the *CC value for the residues and/or include a simulated annealing omit map for the region.

Reply)

As this reviewer suggested, we presented the unbiased difference maps of the H4 N-terminal tails on both sides of the nucleosome (new Extended Data Fig. 2). To obtain this map, the atomic coordinates for the H4 residues 1-25 were deleted from the H3^{CATD} nucleosome. As shown in the new Extended Data Fig. 2, the electron density map for the H4 residues 19-25 is clearly visualized.

Comment 2)

- I would also include a table with the B factors of the different regions including the H4 tail, and other comparing the B factor of the Mn ions/structural waters and the surrounding residues (V76K77).

Reply)

As this reviewer suggested, we presented the plots for the B factors of the H3 and H4 molecules in the H3.1 and CATD nucleosomes in the new Extended Data Fig. 3a. We also presented a table for the B factors of the Mn and oxygen atoms mediating the interaction between the H4 N-terminal tail and the H3Q76 and D77 residues in the new Extended Data Fig. 3b.

Reviewer #3 (Remarks to the Author):

General comment)

In this study, Arimura et al. use X-ray crystal structures of nucleosomes containing WT H3 (H3.1), CENP-A and H3 with the CENP-A centromere targeting domain (CATD) swapped into H3.1 nucleosomes to demonstrate that the H4 tail domain adopts distinct conformations in the crystal structures of these nucleosomes, dependent on two residues in H3.1 (Q76 and D77) that interact with the H4 tail, and constrain its trajectory, while the corresponding residues in the H3 CATD protein (V76 and K77) do not interact with the tail, allowing a more unconstrained structure. The authors hypothesize that the different H4 tail structures explains the localization of monomethylated H4 K20 in CENP-A nucleosomes, suggesting that the residue in H3.1 nucleosomes would be less available for modification. Indeed in vitro methylation assays with the mono-methylase PR-Set7 show a faster methylation of H4 K20 in CENP-A or H3.1 CATD nucleosomes, dependent on V76 and K77. Moreover, ectopic expression of GFP-tagged chicken CENP-A or the CENP-A mutant V67Q/L68D (corresponding to the same residues in the chicken histones) in chicken DT40 cells showed less localization to CEN DNA when ChIP'ed with antibody against monomethylated H4K20 but not with GFP, suggesting monomethylation of CENP-A is dependent on the two CATD residues identified in the structural studies. The data shown is of high quality

and compelling.

The authors conclude that the structural state of the H4 tail, which is defined by interaction with H3.1 (or lack thereof (CENP-A)) in the nucleosome ultimately regulate the activity of the PR-Set7 monomethylase. While an interesting conjecture, I feel that there are some significant questions regarding this conclusion.

Comment 1)

First, could the two residues in question alter the binding of PR-Set7 to the nucleosome? One might imagine that a rather small change in substrate binding free energy, equal to about a single H-bond, could account for the apparent 6-fold change in rate. This would be completely independent of the state of the H4 tail, which is assumed to be quite dynamic. If the enzyme interacts with the CATD domain, this would indeed be a possibility. This is not addressed in the paper, either experimentally or in the discussion.

Reply)

Thank you very much for this insightful comment. To test the dynamic interaction between PR-Set7 and the nucleosomes, we performed gel-shift assays with PR-Set7 and nucleosomes. We then found that the interaction stability between PR-Set7 and the CENP-A nucleosome is substantially weaker than that between PR-Set7 and the H3 nucleosome, suggesting that the PR-Set7 turnover is elevated in the PR-Set7 and CENP-A interaction. We explain that the outward H4-N conformation of the CENP-A nucleosome may allow H4K20 to be a better substrate for PR-Set7, which enhances the catalytic activity in the CENP-A nucleosome, as compared with the H3 nucleosome. These new data are presented in the new Fig. 3f and Extended Data Figure 9, and the results are described in the Results section "CATD stimulates H4K20 monomethylation by PR-Set7 in the nucleosome" of the revised manuscript.

Comment 2)

Second, a major problem with the main conclusion is the fact that the vast majority of H4 in most cells is methylated at K20. Indeed, estimates run from >98% (Pesavento, 2008 MCB doi: 10.1128/MCB.01517-07) to ~85 % in a latter study (Huang, 2015 Chem Rev doi: 10.1021/cr500491u). This, coupled with the fact that PR-Set7 is believed to be the only monomethylase in the cell, and, moreover, the fact that PR-Set7 monomethylation is required for subsequent higher-order methylation events at K20 (K20me2, K20me3) by other enzymes indicates that for the vast majority of nucleosomes (i.e. major H3 nucleosomes) access to the site of modification is not limiting. This makes the proposed model quite untenable. It seems to me that the authors must develop their model within this context, or provide contrary argument. For example, one

possibility, not considered in the MS, is that the CATD domain directly inhibits the installation of additional methyl at K20 groups by Suv4-20.

Reply)

Thank you very much for this insightful comment. I agree that the vast majority of H4K20 is methylated in cells, suggesting that the monomethylation event occurs everywhere in chromosomes, before the conversion of the H4K20 dimethylation or trimethylation of the monomethylated nucleosomes. Interestingly, our previous observations indicated that only small populations of nucleosomes with the dimethylation and trimethylation of H4K20 were detected in the centromere region. In the present study, we demonstrated that the monomethylation is introduced more efficiently into the CENP-A nucleosome than the H3 nucleosome. Considering the limited amounts of dimethylation and trimethylation in centromeres, we propose that the efficient H4K20 monomethylation in the CENP-A nucleosome may cause the restriction of further methylation of the H4K20 residue, which may promote the efficient assembly of other centromeric proteins, including CCAN proteins, in the CENP-A nucleosome. It is also possible that the efficient monomethylation of H4K20 in the CENP-A nucleosomes may confer an advantage for the restriction of the further methylation of H4K20. Of course, further studies will be required for this issue, to clarify our hypothesis. Nevertheless, as we believe that this is an important point, we described our theory in the discussion of the revised manuscript.

Reviewers' comments:

Reviewer #1 (Remarks to the Author):

Arimura et al present a revised version of their manuscript, suggesting a possible role for the CATD of the CENP-A nucleosome in modulating methylation of H4K20 in cis. My principal issues with the original manuscript concerned the structural basis for this mechanism - that an interaction between the Q76 and D77 of histone H3 bind the H4 tail and restrict its conformation. In particular, the proposed interaction between D77 and the main chain carbonyl of L22 seemed dubious. I would like to thank the authors for providing the more detail on this interaction, including release the coordinates of the 3.1 nucleosome. However, on examination, my initial concerns remain. In the case of the D77-L22 interaction, the distance between the Mn²⁺-coordinated water molecule apical to the D77 carboxyl group and the L22 carbonyl oxygen is very large (3.4 angstrom, from extended data figure 6). This is at the very limit of hydrogen bonding distance, which one would expect to be ~2.7 angstrom. Equally, inspecting the side density maps for 5Y0C, the side-chain density of Q76 is very poorly defined, arguing against a stable interaction with R19 as suggested. From looking at the structure, it seems to me to be more plausible that the observed conformation of the H4 tail is driven by an interaction between R19 and the phosphate backbone of dT198.

Taken together, it remains far from clear that there is a particularly strong drive to a given conformation of this tail (the EM maps are pretty weak in this area) and even if there is, it is not clear that interactions with Q76/D77 mediate it. The trypsin accessibility experiment presented showing equal accessibility between H3 and CENP-A nucleosomes would tend to support this. While the data presented does implicate the CATD having some effect on rates of monomethylation, I think that the suggested mechanistic basis is not compelling.

Minor point:

Extended data figure 3. The units of B-factors are Angstroms², not °C.

Reviewer #2 (Remarks to the Author):

While the authors addressed the majority of my points there are still a couple of issues that in my view should be addressed.

The main structural observation in Akimura et al, paper is that both H4 tails adopt an altered conformation in the context of CENP-A residues. I think that it would make sense to include in the Extended Data figure 1 and 2 electron density maps contoured at different (decreasing) sigma levels. The outward configuration might become more clearly visible this way on the disordered side (or as authors call it ambiguous side).

The figures for the cryo-EM structures should be improved, the density of the H4 tails can be showed clearer as this is critical to support their hypothesis

Other than that, the authors should show if the ordered and visible H4 tail that adopts 'outward conformation' in their H3CATD structure is the one that is providing crystal contacts and comment on this in the main text. In canonical H3.1 nucleosomes the H4 tail involved in packing is the one that is ordered and adopts 'inward conformation'.

The HMTase activity showed in the new data is not fully convincing with respect to the difference between the H3CATD and H3.1 nucleosomes. In my opinion, a full enzymatic kinetics is necessary.

The gel shift data has to be repeated with more points so that the apparent K_d 's can be calculated, at this moment it is difficult to judge the binding of PR-Set7. As the outward conformation makes the tail more accessible I would expect tighter binding. While the loss of the affinity resulting in increased enzymatic activity indeed exists in many systems, it doesn't seem to be in agreement with their hypothesis that the H4 tail is more exposed to the methylation in the CENP-A NCP.

Reviewer #3 (Remarks to the Author):

The authors show evidence that two residues in CENP-A (CATD V76 and K77) are responsible for the lack of constraining the H4 tail into an 'inward' orientation, resulting in the 'outward' orientation in crystal structures, which presumably plays a key role in the more efficient monomethylation of H4 K20 by PR-Set7 in CENP-A and H3.1CATD nucleosomes vs H3.1 nucleosomes. The results shown in the paper are compelling and the authors have addressed most of my concerns with several pieces of new data and changes in the text.

One point to be considered is that the authors have not shown that the two residues in question are essential for constraining H4 tail orientation and/or reduced PRSet7 activity in H3.1 nucleosomes, as is implied in several places in the paper. This should be rectified.

Reviewers' comments:

Reviewer #1 (Remarks to the Author):

Comment)

Arimura et al present a revised version of their manuscript, suggesting a possible role for the CATD of the CENP-A nucleosome in modulating methylation of H4K20 in cis. My principal issues with the original manuscript concerned the structural basis for this mechanism - that an interaction between the Q76 and D77 of histone H3 bind the H4 tail and restrict its conformation. In particular, the proposed interaction between D77 and the main chain carbonyl of L22 seemed dubious. I would like to thank the authors for providing the more detail on this interaction, including release the coordinates of the 3.1 nucleosome. However, on examination, my initial concerns remain. In the case of the D77-L22 interaction, the distance between the Mn²⁺-coordinated water molecule apical to the D77 carboxyl group and the L22 carbonyl oxygen is very large (3.4 angstrom, from extended data figure 6). This is at the very limit of hydrogen bonding distance, which one would expect to be ~2.7 angstrom.

Reply)

Thank you for this suggestion. As this reviewer pointed out, the distance between the Mn²⁺-coordinated water molecule and H4L22 is actually 3.4 angstroms, and thus this interaction may be weak. This is now described in the revised manuscript (p.5, ll.28-30).

Comment)

Equally, inspecting the side density maps for 5Y0C, the side-chain density of Q76 is very poorly defined, arguing against a stable interaction with R19 as suggested. From looking at the structure, it seems to me to be more plausible that the observed conformation of the H4 tail is driven by an interaction between R19 and the phosphate backbone of dT198.

Taken together, it remains far from clear that there is a particularly strong drive to a given conformation of this tail (the EM maps are pretty weak in this area)

and even if there is, it is not clear that interactions with Q76/D77 mediate it. The trypsin accessibility experiment presented showing equal accessibility between H3 and CENP-A nucleosomes would tend to support this. While the data presented does implicate the CATD having some effect on rates of monomethylation, I think that the suggested mechanistic basis is not compelling.

Reply)

As this reviewer pointed out, the H4R19 residue directly interacts with the DNA backbone phosphate. Since this H4R19-DNA binding is absent in the H3.1^{CATD} nucleosome, it may contribute to the inward H4-N conformation. However, the H4R19 residue also exists in the H3.1^{CATD} nucleosome. Therefore, there is a mechanism to specifically maintain the H4R19-DNA binding in the H3.1 nucleosome, but not in the H3.1^{CATD} nucleosome. We suspect that, in the H3.1 nucleosome, the H4R19 side-chain orientation may be dictated by the interaction with the H3Q76 residue, assisting the H4R19-DNA binding. This hypothesis is supported by the H3.1^{CATD(V76Q and K77D)} nucleosome structure. In the structure, the electron density of the H3Q76 residue is clearly observed in the location where it could possibly form a hydrogen bond with the H4R19 residue (Extended Data Fig. 6). Therefore, the H3Q76-mediated H4R19-DNA binding may function to maintain the inward H4-N conformation. In the revised manuscript, we presented these H3Q76-H4R19-DNA interactions in the new Extended Data Fig. 6, and discussed this new insight in the Results section “The CATD V76 and K77 residues abrogate the local H3-H4 interaction”.

Minor point:

Comment)

Extended data figure 3. The units of B-factors are Angstroms², not °C.

Reply)

Thank you very much. I corrected it, accordingly.

Reviewer #2 (Remarks to the Author):

Comment 1)

While the authors addressed the majority of my points there are still a couple of issues that in my view should be addressed.

The main structural observation in Akimura et al, paper is that both H4 tails adopt an altered conformation in the context of CENP-A residues. I think that it would make sense to include in the Extended Data figure 1 and 2 electron density maps contoured at different (decreasing) sigma levels. The outward configuration might become more clearly visible this way on the disordered side (or as authors call it ambiguous side). The figures for the cryo-EM structures should be improved, the density of the H4 tails can be showed clearer as this is critical to support their hypothesis

Reply)

To visualize the outward configuration more clearly on the disordered side (ambiguous side), in Extended Data Figures 1 and 2 the electron density maps are shown at different contoured (decreasing) sigma levels, as suggested by this reviewer. We also increased the level of the cryo-EM density maps in Extended Data Figure 4.

Comment 2)

Other than that, the authors should show if the ordered and visible H4 tail that adopts 'outward conformation' in their H3CATD structure is the one that is providing crystal contacts and comment on this in the main text. In canonical H3.1 nucleosomes the H4 tail involved in packing is the one that is ordered and adopts 'inward conformation'.

Reply)

As this reviewer suggested, we commented on the crystal contacts for the H4 N-terminal tail in the H3.1^{CATD} and its mutant nucleosome structures on p.6, ll.13-19.

Comment 3)

The HMTase activity showed in the new data is not fully convincing with respect to the difference between the H3CATD and H3.1 nucleosomes. In my opinion, a full enzymatic kinetics is necessary.

Reply)

We performed the HMTase assay with more precise time course experiments. To do so, substantial amounts of the PR-Set7 protein were required. We purified the recombinant PR-Set7 protein by ourselves (new Extended Data Figure 7c). The freshly prepared PR-Set7 protein exhibited better HMTase activity than the protein supplied by a company. We then re-evaluated all of the HMTase assays in this study with the fresh PR-Set7 protein, and confirmed that the HMTase activities for the H3, H3.1^{CATD}, CENP-A, and CENP-A^{QD} nucleosomes are reproducible with the fresh PR-Set7 protein, although its activity is substantially higher than that of the commercial one. These new data are presented in the new Figure 3d, e, f and the new Extended Data Figure 8, and the results are described in the text (p.7, ll.7-17).

Comment 4)

The gel shift data has to be repeated with more points so that the apparent Kd's can be calculated, at this moment it is difficult to judge the binding of PR-Set7. As the outward conformation makes the tail more accessible I would expect tighter binding. While the loss of the affinity resulting in increased enzymatic activity indeed exists in many systems, it doesn't seem to be in agreement with their hypothesis that the H4 tail is more exposed to the methylation in the CENP-A NCP.

Reply)

Thank you very much for this comment. Using the freshly prepared PR-Set7 protein, we repeated the gel shift assay with more points, and found that PR-Set7 binds to the H3 and CENP-A NCPs with similar affinity. I greatly appreciate this reviewer for giving us a chance to revise this experiment. These

new results are presented in the new Figure 3h and Extended Data Figure 8a, b, and are described in the revised text (p.7, ll.21-22).

Reviewer #3 (Remarks to the Author):

Comment)

The authors show evidence that two residues in CENP-A (CATD V76 and K77) are responsible for the lack of constraining the H4 tail into an 'inward' orientation, resulting in the 'outward' orientation in crystal structures, which presumably plays a key role in the more efficient monomethylation of H4 K20 by PR-Set7 in CENP-A and H3.1CATD nucleosomes vs H3.1 nucleosomes. The results shown in the paper are compelling and the authors have addressed most of my concerns with several pieces of new data and changes in the text.

One point to be considered is that the authors have not shown that the two residues in question are essential for constraining H4 tail orientation and/or reduced PRSet7 activity in H3.1 nucleosomes, as is implied in several places in the paper. This should be rectified.

Reply)

Thank you very much for this comment. I corrected the revised manuscript, according to this suggestion (p.6, ll.17-19 and p.7, ll.14-17).

REVIEWERS' COMMENTS:

Reviewer #2 (Remarks to the Author):

The authors addressed my comments and therefore I support the publication of the paper